# Understanding AdamW through Proximal Methods and Scale-Freeness

## Abstract

Adam has been widely adopted for training deep neural networks due to less hyperparameter tuning and remarkable performance. To improve generalization, Adam is typically used in tandem with a squared $\ell_2$ regularizer (referred to as Adam-$\ell_2$). However, even better performance can be obtained with AdamW, which decouples the gradient of the regularizer from the update rule of Adam-$\ell_2$. Yet, we are still lacking a complete explanation of the advantages of AdamW. In this paper, we tackle this question from both an *optimization* and an *empirical* point of view. First, we show how to re-interpret AdamW as an approximation of a proximal gradient method, which takes advantage of the closed-form proximal mapping of the regularizer instead of only utilizing its gradient information as in Adam-$\ell_2$. Next, we consider the property of "scale-freeness" enjoyed by AdamW and by its proximal counterpart: their updates are invariant to component-wise rescaling of the gradients. We provide empirical evidence across a wide range of deep learning experiments showing a correlation between the problems in which AdamW exhibits an advantage over Adam-$\ell_2$ and the degree to which we expect the gradients of the network to exhibit multiple scales, thus motivating the hypothesis that the advantage of AdamW could be due to the scale-free updates.

## 1 Introduction

Recent years have seen a surge of interest in applying deep neural networks (LeCun et al., 2015) to a myriad of areas. While Stochastic Gradient Descent (SGD) (Robbins & Monro, 1951) remains the dominant method for optimizing such models, its performance depends crucially on the step size hyperparameter which controls how far the algorithm proceeds along the negative stochastic gradient in each step to update the model parameters. To alleviate this problem, people have developed a fruitful line of research on adaptive gradient methods (e.g. Duchi et al., 2010a; McMahan & Streeter, 2010; Tieleman & Hinton, 2012; Zeiler, 2012; Luo et al., 2018; Zhou et al., 2018). These methods provide mechanisms to automatically set stepsizes, and have been shown to greatly reduce the tuning effort while maintaining good performance. Among those adaptive algorithms, one of the most widely used is Adam (Kingma & Ba, 2015) which achieves good results across a variety of problems even by simply adopting the default hyperparameter setting.

In practice, to improve the generalization ability, Adam is typically combined with a $\ell_2$ regularization which adds the squared $\ell_2$ norm of the model weights on top of the loss function (which we will call Adam-$\ell_2$ hereafter). This technique is usually referred to as weight decay because when using SGD, the $\ell_2$ regularization works by first shrinking the model weights by a constant factor in addition to moving along the negative gradient direction in each step. However, as pointed out in Loshchilov & Hutter (2019), for Adam, there is no fixed regularization that achieves this same effect. To address this, they provide a method called AdamW that decouples the gradient of the $\ell_2$ regularization from the update of Adam and directly decays the weights. The two algorithms are shown in Algorithm 1. Although AdamW frequently outperforms Adam-$\ell_2$, the approach is primarily motivated empirically without a clear understanding of why it works so well.

Recently, however, Bjorck et al. (2020) applied AdamW in Natural Language Processing (NLP) and Reinforcement Learning problems and found no improvement of performance over sufficiently tuned Adam-$\ell_2$. Considering the huge popularity of AdamW (Kuen et al., 2019; Lifchitz et al., 2019; Carion et al., 2020), we investigate when and why AdamW has a significant improvement over Adam-$\ell_2$.

In this paper, we focus on understanding the training and testing dynamics of the AdamW update in contrast to Adam-$\ell_2$. We consider this contrast from the lens of optimization theory rather than directly investigating generalization over multiple epochs. First, we unveil the surprising connection between AdamW and *proximal updates*. In particular, we show that AdamW is an approximation of the latter and we also confirm such similarity with an empirical study. Noticing that AdamW and the proximal update are both *scale-free* while Adam-$\ell_2$ is not, we derive a theorem showing that scale-free optimizers enjoy an automatic acceleration with respect to the condition number on certain cases. This gives AdamW a concrete theoretical advantage in training over Adam-$\ell_2$.

Next, we empirically identify the scenario of training very deep neural networks with batch normalization (BN) switched off as a case in which AdamW substantially outperforms Adam-$\ell_2$ in both testing and training. Note that the setting of removing BN is not our invention: indeed, there is already active research in this (De & Smith, 2020; Zhang et al., 2019). The reason is that BN has many disadvantages (Brock et al., 2021) including added memory overhead (Bulò et al., 2018) and training time (Gitman & Ginsburg, 2017), and a discrepancy between training and inferencing (Singh & Shrivastava, 2019). BN has also been found to be not suitable for many cases including distributed computing with a small minibatch per GPU (Wu & He, 2018; Goyal et al., 2017), sequential modeling tasks (Ba et al., 2016), and contrastive learning algorithms (Chen et al., 2020). Moreover, there are already SOTA architectures that do not use BN including the Vision transformer (Dosovitskiy et al., 2021) and the BERT model (Devlin et al., 2019).

For such settings of removing BN, we observe that the magnitudes of the coordinates of the updates during training are much more concentrated about a fixed value for AdamW than for Adam-$\ell_2$, which is an expected property of scale-free algorithms. Further, as depth increases, we expect a greater diversity of gradient scalings, a scenario that should favor scale-free updates. Our experiments support this hypothesis: deeper networks have more dramatic differences between the distributions of update scales between Adam-$\ell_2$ and AdamW, and larger accuracy advantages for AdamW.

Specifically, the contributions of this paper are:

1. We show that AdamW can be seen as an approximation of the proximal updates, which utilize the closed-form proximal mapping of the regularizer instead of only its gradient.
2. We point out the scale-freeness property enjoyed by AdamW and show the advantage of such a property on concrete problem classes.
3. We find a scenario where AdamW is significantly better than Adam-$\ell_2$ in both training and testing performance and report an empirical observation of the correlation between such advantage and the scale-freeness property of AdamW.

The rest of this paper is organized as follows: In Section 2 we discuss the relevant literature. The connection between AdamW and the proximal updates as well as its scale-freeness are explained in Section 3. We then report the empirical observations in Section 4. Finally, we conclude with a discussion of the results and point out some potential future directions.

## 2    RELATED WORK

By enforcing the magnitude of the model weights to be small, weight decay has long been a standard technique to improve the generalization ability in machine learning (Krogh & Hertz, 1991; Bos & Chug, 1996) and is still widely employed in training modern deep neural networks (Devlin et al., 2019; Tan & Le, 2019). Here, we do not attempt to explain the generalization ability of AdamW. Rather, we assume that the regularization and the topology of the network guarantee good generalization performance. Instead, we study algorithms from the perspective of convergence rate.

The use of proximal updates in the batch optimization literature dates back at least to 1965 (Moreau, 1965; Martinet, 1970; Rockafellar, 1976; Parikh & Boyd, 2014), and more recently used even in the stochastic setting (Toulis & Airoldi, 2017; Asi & Duchi, 2019). We are not aware of any previous paper pointing out the connection between AdamW and proximal updates.

The scale-free property was first proposed in the online learning field (Orabona & Pál, 2018). There, they do not need to know a priori the Lipschitz constant bounding the gradient norms while still able to achieve the optimal rates. To the best of our knowledge, scale-freeness has not been explored as an explanation for the efficiency of deep learning optimization algorithms.

**Algorithm 1** Adam with L2 regularization (Adam-$\ell_2$) and Adam with decoupled weight decay (AdamW)
Loshchilov & Hutter (2017) *(Note that these two are exactly the same when $\lambda = 0$, namely no weight decay.)*

1: **Given** $\alpha, \beta_1, \beta_2, \epsilon, \lambda \in \mathbb{R}, \{\eta_t\}_{t \geq 0}$. *All operations on vectors are element-wise.*
2: **Initialize:** $\boldsymbol{x}_0 \in \mathbb{R}^d$, first moment vector $\boldsymbol{m}_0 \leftarrow 0$, second moment vector $\boldsymbol{v}_0 \leftarrow 0$
3: **for** $t = 1, 2, \ldots, T$ **do**
4:     Compute the stochastic gradient $\nabla f_t(\boldsymbol{x}_{t-1})$ evaluated on a mini-batch of samples
5:     $\boldsymbol{g}_t \leftarrow \nabla f_t(\boldsymbol{x}_{t-1}) +\lambda \boldsymbol{x}_{t-1}$
6:     $\boldsymbol{m}_t \leftarrow \beta_1 \boldsymbol{m}_{t-1} + (1 - \beta_1)\boldsymbol{g}_t, \quad \boldsymbol{v}_t \leftarrow \beta_2 \boldsymbol{v}_{t-1} + (1 - \beta_2)\boldsymbol{g}_t^2$
7:     $\hat{\boldsymbol{m}}_t \leftarrow \boldsymbol{m}_t/(1 - \beta_1^t), \quad \hat{\boldsymbol{v}}_t \leftarrow \boldsymbol{v}_t/(1 - \beta_2^t)$
8:     $\boldsymbol{x}_t \leftarrow \boldsymbol{x}_{t-1} -\eta_t\lambda\boldsymbol{x}_{t-1} -\eta_t\alpha\hat{\boldsymbol{m}}_t/(\sqrt{\hat{\boldsymbol{v}}_t} + \epsilon)$
9: **end for**

## 3 THEORETICAL INSIGHTS ON THE MERITS OF ADAMW

**AdamW and Proximal Updates:** Here, we show that AdamW approximates a proximal update with squared $\ell_2$ regularization. This provides a first theoretical motivation for AdamW.

Consider that we want to minimize the objective function

$$F(\boldsymbol{x}) = \tfrac{\lambda}{2}\|\boldsymbol{x}\|_2^2 + f(\boldsymbol{x}), \tag{1}$$

where $\lambda > 0$ and $f(\boldsymbol{x}) : \mathbb{R}^d \rightarrow \mathbb{R}$ is a function bounded from below. We could use a stochastic optimization algorithm that updates in the following fashion

$$\boldsymbol{x}_t = \boldsymbol{x}_{t-1} - M_t\boldsymbol{p}_t, \tag{2}$$

where $M_t$ is a generic matrix containing the learning rates and $\boldsymbol{p}_t$ denotes the update direction. Specifically, we consider $M_t = \eta_t I_d$ where $\eta_t$ is a learning rate schedule, e.g., the constant one or the cosine annealing (Loshchilov & Hutter, 2017). This update covers many cases:

1. $\boldsymbol{p}_t = \boldsymbol{g}_t$ gives us the vanilla SGD;
2. $\boldsymbol{p}_t = \frac{\boldsymbol{g}_t}{\sqrt{\sum_{i=1}^t \boldsymbol{g}_i^2 + \epsilon}}$ gives the AdaGrad algorithm (Duchi et al., 2011);
3. $\boldsymbol{p}_t = \alpha\hat{\boldsymbol{m}}_t/(\sqrt{\hat{\boldsymbol{v}}_t} + \epsilon)$ recovers the Adam algorithm (see Algorithm 1).

Note that in the above we use $\boldsymbol{g}_t$ to denote the stochastic gradient of the entire objective function: $\nabla f_t(\boldsymbol{x}_{t-1}) + \lambda\boldsymbol{x}_{t-1}$ (if the regularizer is not present $\lambda = 0$), with $\hat{\boldsymbol{m}}_t$ and $\hat{\boldsymbol{v}}_t$ both updated using $\boldsymbol{g}_t$.

This update rule (2) is given by the following online mirror descent update (Nemirovsky & Yudin, 1983; Warmuth & Jagota, 1997; Beck & Teboulle, 2003):

$$\boldsymbol{x}_t = \operatorname*{argmin}_{\boldsymbol{x} \in \mathbb{R}^d} \tfrac{\lambda}{2}\|\boldsymbol{x}_{t-1}\|_2^2 + f(\boldsymbol{x}_{t-1}) + \boldsymbol{p}_t^\top(\boldsymbol{x} - \boldsymbol{x}_{t-1}) + \tfrac{1}{2}(\boldsymbol{x} - \boldsymbol{x}_{t-1})^\top M_t^{-1}(\boldsymbol{x} - \boldsymbol{x}_{t-1}).$$

This approximates minimizing a first-order Taylor approximation of $F$ centered in $\boldsymbol{x}_{t-1}$ plus a term that measures the distance between the $\boldsymbol{x}_t$ and $\boldsymbol{x}_{t-1}$ according to the matrix $M_t^{-1}$. The approximation becomes exact when $\boldsymbol{p}_t = \nabla f(\boldsymbol{x}_{t-1}) + \lambda\boldsymbol{x}_{t-1}$.

However, this is not the only way to construct first-order updates for the objective function (1). An alternative route is to linearize only $f$ and to keep the squared $\ell_2$ norm in its function form:

$$\boldsymbol{x}_t = \operatorname*{argmin}_{\boldsymbol{x} \in \mathbb{R}^d} \tfrac{\lambda}{2}\|\boldsymbol{x}\|_2^2 + f(\boldsymbol{x}_{t-1}) + \boldsymbol{p}_t^\top(\boldsymbol{x} - \boldsymbol{x}_{t-1}) + \tfrac{1}{2}(\boldsymbol{x} - \boldsymbol{x}_{t-1})^\top M_t^{-1}(\boldsymbol{x} - \boldsymbol{x}_{t-1}),$$

This update rule is using the *proximal operator* (Moreau, 1965; Parikh & Boyd, 2014) of $\frac{1}{2}\|\cdot\|_2^2$ with respect to the norm $\|\cdot\|_{M_t^{-1}}$. It is intuitive why this would be a better update: *We directly minimize the squared $\ell_2$ norm instead of approximating it.* From the first-order optimality condition, we have

$$\boldsymbol{x}_t = (I_d + \lambda M_t)^{-1}(\boldsymbol{x}_{t-1} - M_t\boldsymbol{p}_t). \tag{3}$$

When $\lambda = 0$, the update in (2) and this one coincide. Yet, when $\lambda \neq 0$, they are no longer the same.

We now show how the update in (3) generalizes the one in AdamW. The update of AdamW is

$$\boldsymbol{x}_t = (1 - \lambda\eta_t)\boldsymbol{x}_{t-1} - \eta_t\alpha\hat{\boldsymbol{m}}_t/(\sqrt{\hat{\boldsymbol{v}}_t} + \epsilon) . \tag{4}$$

On the other hand, using $M_t = \eta_t I_d$ and $\boldsymbol{p}_t = \alpha\hat{\boldsymbol{m}}_t/(\sqrt{\hat{\boldsymbol{v}}_t} + \epsilon)$ in (3) gives:

$$\boldsymbol{x}_t = (1 + \lambda\eta_t)^{-1}(\boldsymbol{x}_{t-1} - \eta_t\alpha\hat{\boldsymbol{m}}_t/(\sqrt{\hat{\boldsymbol{v}}_t} + \epsilon)), \tag{5}$$

which we will call *AdamProx* afterward. Its first-order Taylor approximation around $\eta_t = 0$ is

$$\boldsymbol{x}_t \approx (1 - \lambda\eta_t)\boldsymbol{x}_{t-1} - \eta_t\alpha\hat{\boldsymbol{m}}_t/(\sqrt{\hat{\boldsymbol{v}}_t} + \epsilon),$$

which is exactly the AdamW update (4) as in the original paper.

The careful reader might notice that the approximation of AdamW to AdamProx becomes less accurate when $\eta_t$ becomes too large, and so be concerned whether this approximation is practical at all. Fortunately, in practice, $\eta_t$ is never large enough for this to be an issue. Most practical learning rate schedules, e.g., cosine, exponential, polynomial, and step decay, all decrease from $\eta_0 = 1$, or some even smaller value. Thus, the remainder term of this approximation is $O(\lambda\eta_t^2) \leq O(\lambda\eta_0^2)$ which we should always expect to be small as both $\lambda$ and $\eta_t$ are small. Consequently, we can expect AdamW and AdamProx to perform similarly for learning rate schedules $\eta_t$ commonly employed in practice, and we will indeed confirm this empirically in Section 4.3.

While perhaps not widely known, the theory of proximal methods is a deep and beautiful branch of optimization (see Parikh & Boyd (2014) for a survey). On the other hand, although AdamW is a very popular practical algorithm (e.g., for training BERT (Devlin et al., 2019) and vision transformer-based architectures (Dosovitskiy et al., 2021)), it is still unclear how it works so well. So, this connection opens the door to new ways to design optimization algorithms. For example, the proximal update gives an immediate theoretical advantage: the convergence rate, at least in the convex case, will depend on $\|\nabla f(\boldsymbol{x}_t)\|^2_{M_t}$ rather than on $\|\nabla f(\boldsymbol{x}_t) + \lambda\boldsymbol{x}_t\|^2_{M_t}$ Duchi et al. (2010b).

**AdamW is Scale-Free** An optimization algorithm is said to be *scale-free* if its iterates do not change when one multiplies any coordinate of all the gradients by a positive constant (Orabona & Pál, 2018). It turns out that the update (4) of AdamW and the update (5) of AdamProx are both scale-free when $\epsilon = 0$. This is evident for AdamW, since the scaling factor for any coordinate of the gradient is kept in both $\hat{\boldsymbol{m}}_t$ and $\sqrt{\hat{\boldsymbol{v}}_t}$ and will be canceled out when dividing them. In contrast, for Adam-$\ell_2$, the addition of the weight decay vector to the gradient (see Line 5 of Algorithm 1) destroys this property. Note that in practical applications $\epsilon$ is very small but not zero, yet we empirically verify in Section 4.2 that it is small enough to still approximately guarantee the scale-free property.

We want to emphasize the comparison between Adam-$\ell_2$ and AdamW: once Adam-$\ell_2$ adopts non-zero $\lambda$, it loses the scale-freeness property; in contrast, AdamW still enjoys this for arbitrary $\lambda$. The same applies to any AdaGrad-type and Adam-type algorithms that employ weight decay strategy in the same way as Adam-$\ell_2$ by simply adding the gradient of the $\ell_2$ regularizer directly to the gradient of $f$ (as implemented in Tensorflow and Pytorch). Such algorithms are scale-free only when they do not use weight decay. In fact, this is one of our main observations: scale-freeness helps optimizers in deep learning and AdamW preserves this scale-freeness property even with an $\ell_2$ regularizer.

We stress that the scale-freeness is an important but largely overlooked property of an optimization algorithm. It has already been utilized to explain the success of AdaGrad (Orabona & Pál, 2018). Recently, Agarwal et al. (2020) also advocates for setting the $\epsilon$ in the denominator of AdaGrad to be 0 thus making the update indeed scale-free, and they also provide an NLP experiment where such choice yields the best performance.

Now, we show the effect of scale-freeness from a theoretical point of view. Consider a twice continuously differentiable function $f$. It is well-known that the convergence rate of many optimization algorithms, like gradient descent, on minimizing $f$ depends on the condition number $\kappa(\nabla^2 f(x))$, i.e., the ratio of its largest eigenvalue to its smallest eigenvalue of the Hessian (see, e.g. Nesterov, 2004). It turns out that scale-freeness can effectively reduce the effects of the condition number, as detailed by the following theorem whose proof can be found in the appendix.

**Theorem 1.** *Let $f$ be a twice continuously differentiable function such that $x^* \in \arg\min f(x)$. Then, let $\tilde{f}_\Lambda$ be the family of functions such that $x^* \in \arg\min \tilde{f}_\Lambda(x)$, and $\nabla^2 \tilde{f}_\Lambda(x) = \Lambda\nabla^2 f(x)$, where $\Lambda = diag(\lambda_1, \ldots, \lambda_d) \succeq 0$. Consider a scale-free algorithm whose convergence rate can be bounded depending on the condition number of the objective function. Then, the convergence of such algorithm used to minimize $f$ will only depend on the smallest condition number among all functions $\tilde{f}_\Lambda$.*

To give an example of when this is advantageous, consider when $\nabla^2 f(x)$ is a diagonal matrix:

$$\nabla^2 f(x) = diag(g_1(x), g_2(x), \ldots, g_d(x)) \ .$$

Assume $\mu \leq \mu_i \leq g_i(x) \leq M_i \leq M$ for $i \in \{1, \ldots, d\}$. Denote $j = \arg\max_i M_i/\mu_i$. Choose $\lambda_i$ s.t. $\mu_j \leq \lambda_i \mu_i \leq \lambda_i g_i(x) \leq \lambda_i M_i \leq M_j$ and we have that $\Lambda \nabla^2 f(x)$ has a condition number $\kappa' = M_j/\mu_j$. This gives scale-free algorithms a big advantage when $\max_i M_i/\mu_i \ll M/\mu$. Specifically:

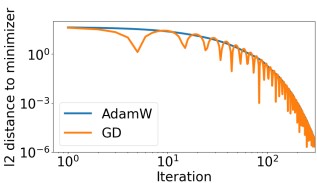

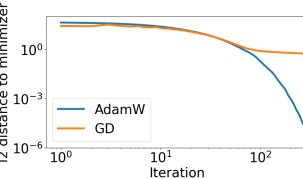

(a) Condition number 1.

(b) Condition number 100000.

Figure 1: Non-scale-free GD v.s. scale-free AdamW on quadratic functions with different condition numbers.

**Corollary 1.** *For quadratic problems $f(x) = \frac{1}{2} x^\top H x + b^\top x + c$, with $H$ diagonal and positive definite, any scale-free algorithm will not differentiate between minimizing $f$ and $\tilde{f}(x) = \frac{1}{2} x^\top x + H^{-1} b^\top x + c$. As the condition number of $\tilde{f}$ is 1, the operation, and most importantly, the convergence, of a scale-free algorithm will not be affected by the condition number of $f$ at all.*

Figure 1 illustrates Corollary 1. Here, we compare GD (non-scale-free) with AdamW (scale-free) on optimizing two quadratic functions with the same minimizer, but one hessian matrix being a rescaled version of the other, resulting in different condition numbers. Even after tuning the learning rates, the figure clearly shows that the updates of AdamW (starting from the same point) and thus its convergence to the minimizer, is completely unaffected by the condition number, while GD's updates change drastically and its performance deteriorates significantly when the condition number is large. Moreover, note that poor training performance mostly likely will also lead to poor testing performance.

This also explains AdaGrad's improvements over SGD in certain scenarios. As an example, in Appendix B we analyze a variant of AdaGrad with restarts and show an improved convergence on strongly convex functions due to scale-freeness. Note that the folklore justification for such improvements is that the learning rate of AdaGrad approximates the inverse of the Hessian matrix, but this is incorrect: AdaGrad does not compute Hessians and there is no reason to believe it approximates them in general.

More importantly, another scenario demonstrating the advantage of scale-freeness is training deep neural networks. Neural networks are known to suffer from the notorious problem of vanishing/exploding gradients (Bengio et al., 1994; Glorot & Bengio, 2010; Pascanu et al., 2013). This problem leads to the gradient scales being very different across layers, especially between the first and the last layers. The problem is particularly severe when the model is not equipped with normalization mechanisms like Batch Normalization (BN) (Ioffe & Szegedy, 2015). In such cases, when using a non-scale-free optimization algorithm (e.g. SGD), the first layers and the last layers will proceed at very different speeds, whereas a scale-free algorithm ensures that each layer is updated at a similar pace.

## 4 EMPIRICAL FINDINGS OF ADAMW ON DEEP LEARNING EXPERIMENTS

In this section, we empirically compare Adam-$\ell_2$ with AdamW. First (Section 4.1), we report experiments for deep neural networks on image classification tasks (CIFAR10/100). Here, AdamW enjoys a significant advantage over Adam-$\ell_2$ when the batch normalization is switched off on deeper neural networks. We also report the correlation between this advantage and the scale-freeness property of AdamW. Next (Section 4.2), we show that AdamW is still almost scale-free even when the $\epsilon$ used in practice is not 0, and how, contrary to AdamW, Adam-$\ell_2$ is not invariant of the loss function being multiplied by a positive constant. Finally (Section 4.3), we show that AdamW performs similarly to AdamProx in practice, thus supporting the observations in Section 3.

**Data Normalization and Augmentation:** We consider the image classification task on CIFAR-10/100 datasets. Images are normalized per channel using the means and standard deviations computed from all training images. We adopt the data augmentation technique following Lee et al. (2015) (for training only): 4 pixels are padded on each side of an image and a $32 \times 32$ crop is randomly sampled from the padded image or its horizontal flip.

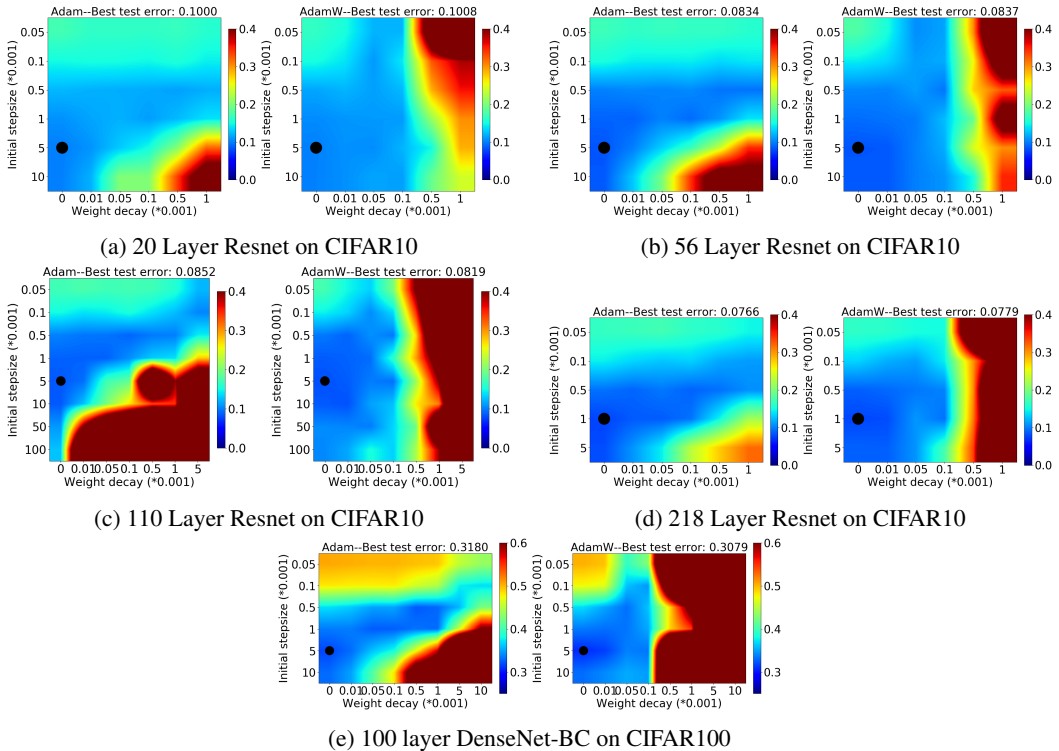

Figure 2: The final Top-1 test error on using AdamW vs. Adam-$\ell_2$ on training a Resnet/DenseNet with Batch Normalization on CIFAR10/100 (*where the black circle denotes the best setting*).

**Models:** For the CIFAR-10 dataset, we employ the Residual Network[1] model (He et al., 2016) of 20/44/56/110/218 layers; and for CIFAR-100, we utilize the DenseNet-BC[2] model (Huang et al., 2017) with 100 layers and a growth-rate of 12. The loss is the cross-entropy loss.

**Hyperparameter tuning:** For both Adam-$\ell_2$ and AdamW, we set $\beta_1 = 0.9$, $\beta_2 = 0.999$, $\epsilon = 10^{-8}$ as suggested in the original Adam paper Kingma & Ba (2015). To set the initial step size $\alpha$ and weight decay parameter $\lambda$, we grid search over $\{0.00005, 0.0001, 0.0005, 0.001, 0.005\}$ for $\alpha$ and $\{0, 0.00001, 0.00005, 0.0001, 0.0005, 0.001\}$ for $\lambda$. Whenever the best performing hyperparameters lie in the boundary of the searching grid, we always extend the grid to ensure that the final best-performing hyperparameters fall into the interior of the grid.

**Training:** For each experiment configuration (e.g. 110-layer Resnet batch normalization), we randomly select an initialization of the model to use as a fixed starting point for all optimizers and hyperparameter settings. We use a mini-batch of 128, and train 300 epochs unless otherwise specified.

## 4.1 ADAMW VS. ADAM-$\ell_2$ ON IMAGE CLASSIFICATION TASKS, THE INFLUENCE OF BATCH NORMALIZATION, AND THE CORRELATION WITH SCALE-FREENESS

**With BN, Adam-$\ell_2$ is on par with AdamW:** Recently, Bjorck et al. (2020) found that AdamW has no improvement in absolute performance over sufficiently tuned Adam-$\ell_2$ in some reinforcement learning experiments. We also discover the same phenomenon in several image classification tasks, see Figure 2. Indeed, the best weight decay parameter is 0 for all cases and AdamW coincides with Adam-$\ell_2$ in these cases. Nevertheless, AdamW does decouple the optimal choice of the weight decay parameter from the initial step size much better than Adam-$\ell_2$ in all cases.

**Removing BN:** Notice that the models used in Figure 2 all employ Batch Normalization (BN) (Ioffe & Szegedy, 2015). BN works by normalizing the input to each layer across the mini-batch to make each

---

[1]https://github.com/akamaster/pytorch_resnet_cifar10
[2]https://github.com/bearpaw/pytorch-classification

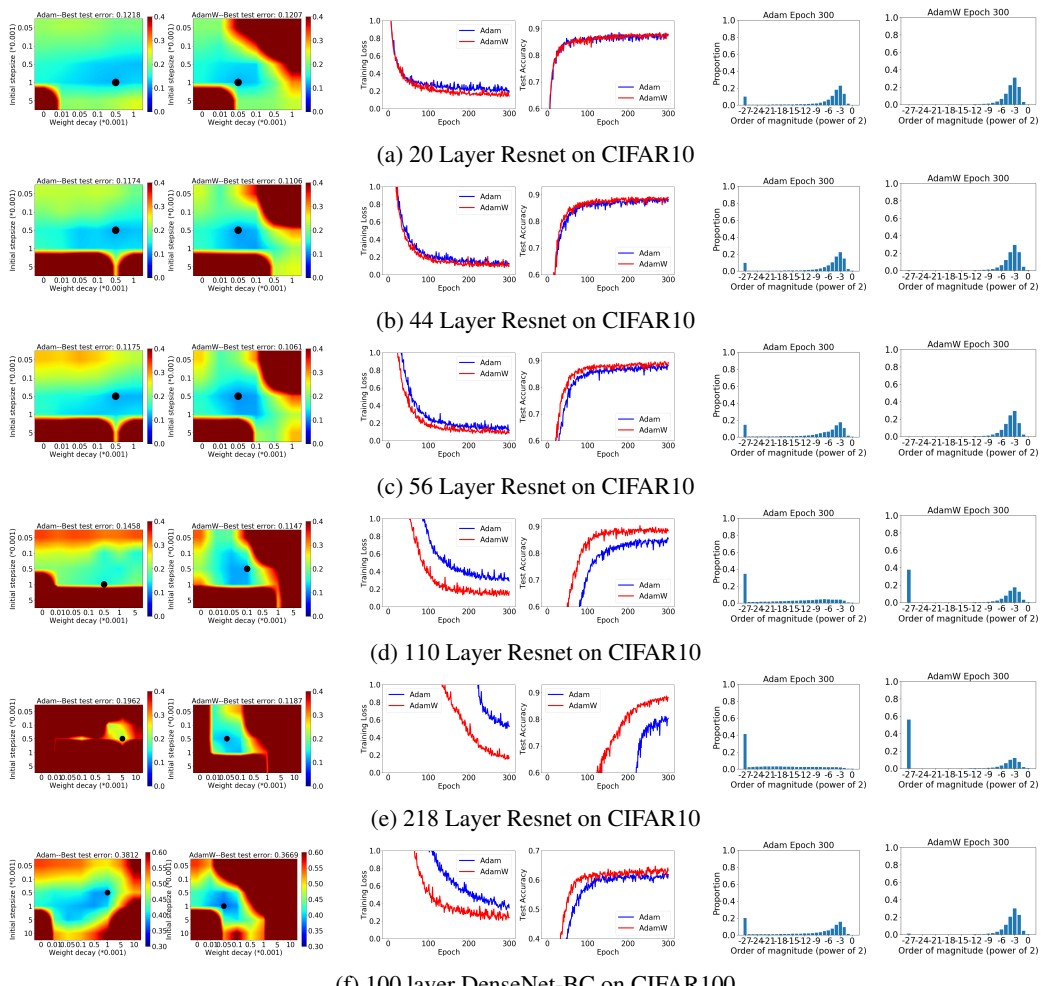

Figure 3: On using AdamW vs. Adam-$\ell_2$ on training a Resnet/DenseNet without Batch Normalization on CIFAR10/100. (Left two) The final Top-1 test error (*where the black circle denotes the best setting*). (Middle two) The training loss and test accuracy curve when employing the initial step size and the weight decay parameter that gives the smallest test error. (Right two) The histogram of the magnitude of corresponding updates of all coordinates of the network near the end of the training when employing the initial step size and the weight decay parameter that gives the smallest test error.

coordinate have zero-mean and unit-variance. Without BN, deep neural networks are known to suffer from gradient explosion and vanishing (Schoenholz et al., 2017). This means each coordinate of the gradient will have very different scales, especially between the first and last layers. For non-scale-free algorithms, the update to the network weights will also be affected and each coordinate will proceed at a different pace. In contrast, scale-free optimizers are robust to such issues as the scaling of any single coordinate will not affect the update. Also, we already mentioned in the introduction some other reasons for considering removing BN. Thus, we consider the case where BN is removed as that is where AdamW and Adam-$\ell_2$ will show very different patterns due to scale-freeness.

**Without BN, AdamW Outperforms Adam-$\ell_2$:** In fact, without BN, AdamW outperforms Adam-$\ell_2$ even when both are finely tuned, especially on relatively deep neural networks (see Figure 3). AdamW not only obtains a much better test accuracy but also trains much faster.

**Correlation between AdamW's Advantage and Scale-freeness:** We also observe that the advantage of AdamW becomes more evident as the network becomes deeper. Recall that as the depth grows, without BN, the gradient explosion and vanishing problem will become more severe. This means that for the non-scale-free Adam-$\ell_2$, the updates of each coordinate will be dispersed on a wider range of

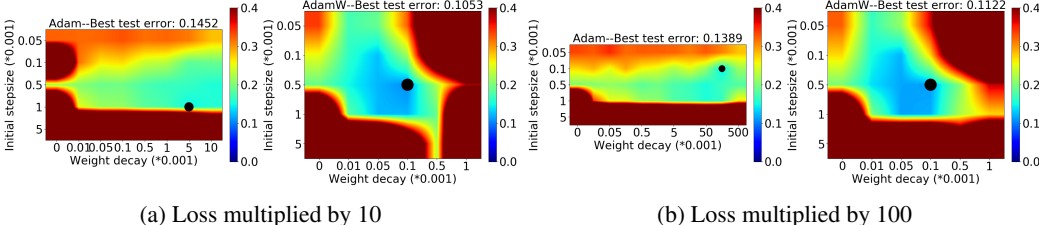

(a) Loss multiplied by 10     (b) Loss multiplied by 100

Figure 4: The final top-1 test error of AdamW vs. Adam-$\ell_2$ on optimizing a 110-layer Resnet with batch normalization *removed* optimized by AdamW or Adam with the loss function multiplied by 10 (left two figures) and 100 (right two figures).

scales even when the same weight decay parameter is employed. In contrast, the scales of the updates of AdamW will be much more concentrated in a smaller range. This is exactly verified empirically as illustrated in the 5th & 6th columns of figures in Figure 3. There, we report the histograms of the absolute value of updates of Adam-$\ell_2$ vs. AdamW of all coordinates near the end of training (for their comparison during the whole training process please refer to the Appendix).

This correlation between the advantage of AdamW over Adam-$\ell_2$ and the different spread of corresponding update scales which is induced by the scale-freeness property of AdamW provides an empirical justification on when and why AdamW excels over Adam-$\ell_2$.

**SGD and Scale-freeness:** The reader might wonder why SGD is known to provide state-of-the-art performance on many deep learning architectures (e.g., He et al., 2016; Huang et al., 2017) *without* being scale-free. At first blush, this seems to contradict our claims that scale-freeness correlates with good performance. In reality, the good performance of SGD in very deep models is linked to the use of BN that normalizes the gradients. Indeed, we verified empirically that SGD fails spectacularly when BN is not used. For example, on training the 110 layer Resnet without BN using SGD with momentum and weight decay of $0.0001$, even a learning rate of $1e-10$ will lead to divergence.

## 4.2 Verifying Scale-Freeness: AdamW vs. Adam-$\ell_2$ under Loss Multiplication

In the previous section, we elaborated on the scale-freeness property of AdamW and its correlation with AdamW's advantage over Adam-$\ell_2$. However, one may notice that in practice, the $\epsilon$ factor in the AdamW update is typically small but not 0, in our case $1e$-8, thus preventing it from completely scale-free. In this section, we verify that the effect of such an $\epsilon$ on the scale-freeness is negligible.

As a simple empirical verification of the scale-freeness, we consider the scenario where we multiply the loss function by a positive number. Note that any other method to test scale-freeness would be equally good. For a feed-forward neural network without BN, this means the gradient would also be scaled up by that factor. In this case, the updates of a scale-free optimization algorithm would remain exactly the same, whereas they would change for an optimization algorithm that is not scale-free.

Figure 4 shows results of the loss function being multiplied by 10 and 100 respectively on optimizing a 110-layer Resnet with BN *removed*. For results of the original loss see Figure 3d. We can see that AdamW has almost the same performance across the range of initial learning rates and weight decay parameters, and most importantly, the best values of these two hyperparameters remain the same. This verifies that, even when employing a (small) non-zero $\epsilon$, AdamW is still approximately scale-free. In contrast, Adam-$\ell_2$ is not scale-free and we can see that its behavior varies drastically with the best initial learning rates and weight decay parameters in each setting totally different.

## 4.3 AdamW and AdamProx Behave very Similarly in Practice

In Section 3, we showed theoretically that AdamW is the first order Taylor approximation of AdamProx (update rule (5)). Beyond this theoretical argument, we also verified empirically that the approximation is good. In Figure 5, we consider the case when $\eta_t = 1$ for all $t$ - a relatively large constant learning rate schedule. In such cases, AdamW and AdamProx still behave very similarly. This suggests that for most learning rate schedules, e.g., cosine, exponential, polynomial, and step decay, which all monotonously decrease from $\eta_0 = 1$, AdamProx will remain a very good approximation to AdamW. Thus, it is reasonable to try to understand AdamW by instead understanding the more classically-linked AdamProx.

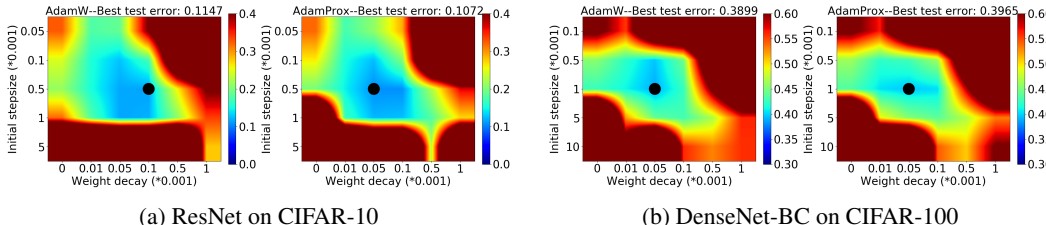

(a) ResNet on CIFAR-10        (b) DenseNet-BC on CIFAR-100

Figure 5: The final Top-1 test error of using AdamW vs. AdamProx on training (*where the black circle denotes the best setting*). (Top row) a 110-layer ResNet with Batch-normalization *removed* on CIFAR-10 (trained for 300 epochs). (Bottom row) a 100-layer DenseNet-BC with Batch-normalization *removed* on CIFAR-100 (trained for 100 epochs).

## 5 CONCLUSION AND FUTURE WORKS

In this paper, we provide explanations for the merits of AdamW from two points of view. We first show that AdamW is an approximation of the proximal updates both theoretically and empirically. We then identify the setting of training very deep neural networks without batch normalization in which AdamW substantially outperforms Adam-$\ell_2$ in both training and testing, and provide an empirical explanation through the scale-freeness property of AdamW. Nevertheless, there are still many directions worth exploring and we name a few here.

**Update for no-square $\ell_2$ regularization.** Instead of using the squared $\ell_2$ regularization, we might think to use the $\ell_2$ regularization, that is without the square. This is known to have better statistical properties than the squared $\ell_2$ (Orabona, 2014), but it is not smooth so it is harder to be optimized. However, with proximal updates, we don't have to worry about its non-smoothness. Hence, we can consider the objective function

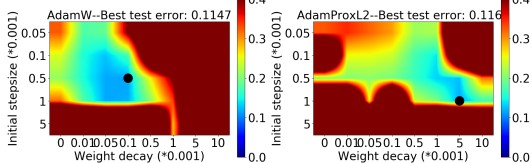

Figure 6: The final Top-1 test error of using AdamW vs. AdamProxL2 to train a 110-layer ResNet with BN *removed* on CIFAR10 (*where the black circle denotes the best setting*).

$$F(\boldsymbol{x}) = \lambda\|\boldsymbol{x}\|_2 + f(\boldsymbol{x}) .$$

The corresponding prox-SGD update was derived in Duchi & Singer (2009) for scalar learning rates and it is easy to see that it generalizes to our setting for $M_t = \eta_t I_d$ as

$$\boldsymbol{x}_{t+1} = \max\left(1 - \frac{\lambda\eta_t}{\|\boldsymbol{x}_t - \eta_t\boldsymbol{p}_t\|}, 0\right)(\boldsymbol{x}_t - \eta_t\boldsymbol{p}_t) .$$

Its performance, named AdamProxL2, as shown in Figure 6, can be on a par with AdamW.

**Update for $\ell_1$ regularization.** The $\ell_1$ regularization is widely used for achieving the sparsity of the solution. Recently, Neyshabur (2020) proposed $\beta$-lasso, a variant of the LASSO algorithm or namely regression with $\ell_1$ regularization, and showed significant improvements on training fully-connected networks. Thus, we are curious to see how the variant of AdamW derived from using $\ell_1$ regularization instead will behave. Specifically, we consider the objective function:

$$F(\boldsymbol{x}) = \lambda\|\boldsymbol{x}\|_1 + f(\boldsymbol{x}),$$

for which the proximal update is obtained by solving:

$$\underset{\boldsymbol{x}}{\operatorname{argmin}} \tfrac{1}{2}[(\boldsymbol{x} - (\boldsymbol{x}_t - M_t\boldsymbol{p}_t)]^T M_t^{-1}[\boldsymbol{x} - (\boldsymbol{x}_t - M_t\boldsymbol{p}_t)] + \lambda\|\boldsymbol{x}\|_1 .$$

When $M_t$ is a diagonal matrix with positive eigenvalues, the above objective function is decomposable and we can solve for each dimension. The solution for 1-d case was derived in Duchi & Singer (2009) and it is easy to see that it generalizes to our setting as

$$\boldsymbol{x}_{t+1,i} = \operatorname{sign}(\boldsymbol{x}_t - M_t\boldsymbol{p}_t)_i \max(|\boldsymbol{x}_t - M_t\boldsymbol{p}_t|_i - \lambda M_{t,i}, 0) .$$

**Distributed Training** As already noted in Section 4.1, batch normalization is not very compatible with distributed training. Since AdamW outperforms Adam-$\ell_2$ significantly in settings with BN switched off, at least in feed-forward neural networks, we can apply AdamW in distributed training to see if it still enjoys the same merits.

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

# Appendices

## A   PROOF OF THEOREM 1

*Proof.* From the Fundamental Theorem of Calculus we have:

$$\nabla f(x) = \nabla f(x^*) + \int_0^1 \nabla^2 f(x^* + t(x - x^*))(x - x^*)dt = \int_0^1 \nabla^2 f(x^* + t(x - x^*))(x - x^*)dt .$$

Thus, for any function $\tilde{f}_\Lambda(x)$ whose Hessian is $\Lambda \nabla^2 f(x)$ and $\nabla \tilde{f}_\Lambda(x^*) = 0$, $\nabla \tilde{f}_\Lambda(x) = \Lambda \nabla f(x)$.

Now, from the definition of a scale-free algorithm, the iterates of such an algorithm do not change when one multiplies any coordinate of all the gradients by a positive constant. Thus, a scale-free algorithm optimizing $f$ behaves the same as if it is optimizing $\tilde{f}_\Lambda$. □

## B   A SCALE-FREE ALGORITHM WITH DEPENDENCY ON THE CONDITION NUMBER

---

**Algorithm 2** AdaGrad (Duchi et al., 2011; McMahan & Streeter, 2010) *(All operations on vectors are element-wise.)*

---

**Input**: #Iterations $T$, a set $\mathcal{K}$, $\boldsymbol{x}_1 \in \mathcal{K}$, stepsize $\eta$
**for** $t = 1 \ldots T$ **do**
  Receive: $\nabla f(\boldsymbol{x}_t)$
  Set: $\boldsymbol{\eta}_t = \frac{\eta}{\sqrt{\sum_{i=1}^t (\nabla f(\boldsymbol{x}_i))^2}}$
  Update: $\boldsymbol{x}_{t+1} = \Pi_{\mathcal{K}}(\boldsymbol{x}_t - \boldsymbol{\eta}_t \nabla f(\boldsymbol{x}_t))$ where $\Pi_{\mathcal{K}}$ is the projection onto $\mathcal{K}$.
**end for**
Output: $\bar{\boldsymbol{x}} = \frac{1}{T} \sum_{t=1}^T \boldsymbol{x}_t$.

---

---

**Algorithm 3** AdaGrad with Restart

---

**Input**: #Rounds $N$, $\boldsymbol{x}_0 \in \mathbb{R}^d$, upper bound on $\|\boldsymbol{x}_0 - \boldsymbol{x}^*\|_\infty$ as $D_\infty$, strong convexity $\mu$, smoothness $M$
Set: $\bar{\boldsymbol{x}}_0 = \boldsymbol{x}_0$
**for** $i = 1 \ldots N$ **do**
  Run Algorithm 2 to get $\bar{\boldsymbol{x}}_i$ with $T = 32d\frac{M}{\mu}$, $\boldsymbol{x}_1 = \bar{\boldsymbol{x}}_{i-1}$, $\mathcal{K} = \{\boldsymbol{x} : \|\boldsymbol{x} - \bar{\boldsymbol{x}}_{i-1}\|_\infty^2 \leq \frac{D_\infty^2}{4^{i-1}}\}$, $\eta = \frac{D_\infty / \sqrt{2}}{2^{i-1}}$
**end for**
Output: $\bar{\boldsymbol{x}}_N$.

---

**Theorem 2.** *Let $\mathcal{K}$ be a hypercube with $\|\boldsymbol{x} - \boldsymbol{y}\|_\infty \leq D_\infty$ for any $\boldsymbol{x}, \boldsymbol{y} \in \mathcal{K}$. For a convex function $f$, set $\eta = \frac{D_\infty}{\sqrt{2}}$, then Algorithm 2 guarantees for any $\boldsymbol{x} \in \mathcal{K}$:*

$$\sum_{t=1}^T f(\boldsymbol{x}_t) - f(\boldsymbol{x}) \leq \sqrt{2dD_\infty^2 \sum_{t=1}^T \|\nabla f(\boldsymbol{x}_t)\|^2} . \tag{6}$$

**Theorem 3.** *For a $\mu$ strongly convex and $M$ smooth function $f$, denote its unique minimizer as $\boldsymbol{x}^* \in \mathbb{R}^d$. Given $\boldsymbol{x}_0 \in \mathbb{R}^d$, assume that $\|\boldsymbol{x}_0 - \boldsymbol{x}^*\|_\infty \leq D_\infty$, then Algorithm 3 guarantees:*

$$\|\bar{\boldsymbol{x}}_N - \boldsymbol{x}^*\|_\infty^2 \leq \frac{D_\infty^2}{4^N} .$$

*Thus, to get a $\boldsymbol{x}$ such that $\|\boldsymbol{x} - \boldsymbol{x}^*\|_\infty^2 \leq \epsilon$, we need at most $32d\frac{M}{\mu} \log_4\left(D_\infty^2/\epsilon\right)$ gradient calls.*

*Proof of Theorem 3.* Consider round $i$ and assume $\mathcal{K}$ passed to Algorithm 2 is bounded w.r.t. $\ell_\infty$ norm by $D_{\infty_i}$. When $f$ is $\mu$-strongly convex and $M$ smooth, let $\boldsymbol{x} = \boldsymbol{x}^*$, Equation (6) becomes:

$$\sum_{t=1}^T f(\boldsymbol{x}_t) - f(\boldsymbol{x}^*) \leq \sqrt{2dD_{\infty_i}^2 \sum_{t=1}^T \|\nabla f(\boldsymbol{x}_t)\|^2} \leq \sqrt{4MdD_{\infty_i}^2 \sum_{t=1}^T (f(\boldsymbol{x}_t) - f(\boldsymbol{x}^*))} ,$$

where the second inequality is by the $M$ smoothness of $f$. This gives:

$$\sum_{t=1}^{T} f(\boldsymbol{x}_t) - f(\boldsymbol{x}^*) \le 4MdD_{\infty_i}^2 .$$

Let $\bar{\boldsymbol{x}}_i = \frac{1}{T} \sum_{t=1}^{T} \boldsymbol{x}_t$ we have by the $\mu$-strong-convexity that:

$$\|\bar{\boldsymbol{x}}_i - \boldsymbol{x}^*\|_{\infty}^2 \le \|\bar{\boldsymbol{x}}_i - \boldsymbol{x}^*\|^2 \le \frac{2}{\mu}(f(\bar{\boldsymbol{x}}) - f(\boldsymbol{x}^*)) \le \frac{2}{\mu}\frac{1}{T}\sum_{t=1}^{T}(f(\boldsymbol{x}_t) - f(\boldsymbol{x}^*)) \le \frac{8MdD_{\infty_i}^2}{\mu T} . \quad (7)$$

Put $T = 32d\frac{M}{\mu}$ in Equation (7) we have that $\|\bar{\boldsymbol{x}}_i - \boldsymbol{x}^*\|_{\infty}^2 \le \frac{D_{\infty_i}^2}{4}$. Thus, after each round, the $\ell_{\infty}$ distance between the update $\bar{\boldsymbol{x}}_i$ and $\boldsymbol{x}^*$ is shrinked by half, which in turn ensures that $\boldsymbol{x}^*$ is still inside the $\mathcal{K}$ passed to Algorithm 2 in the next round with $D_{\infty_{i+1}} = \frac{D_{\infty_i}}{2}$. This concludes the proof. $\quad\square$

*Proof of Theorem 2.*

$$\sum_{t=1}^{T} f(\boldsymbol{x}_t) - f(\boldsymbol{x})$$

$$\le \sum_{t=1}^{T} \langle \nabla f(\boldsymbol{x}_t), \boldsymbol{x}_t - \boldsymbol{x} \rangle$$

$$= \sum_{t=1}^{T}\sum_{j=1}^{d} \frac{\partial f}{\partial x_{t,j}}(\boldsymbol{x}_t) * (x_{t,j} - x_j)$$

$$= \sum_{t=1}^{T}\sum_{j=1}^{d} \frac{(x_{t,j} - x_j)^2 - \left(x_{t,j} - \eta_{t,j}\frac{\partial f}{\partial x_{t,j}}(\boldsymbol{x}_t) - x_j\right)^2}{2\eta_{t,j}} + \sum_{t=1}^{T}\sum_{j=1}^{d}\frac{\eta_{t,j}}{2}\left(\frac{\partial f}{\partial x_{t,j}}(\boldsymbol{x}_t)\right)^2$$

$$\le \sum_{t=1}^{T}\sum_{j=1}^{d} \frac{(x_{t,j} - x_j)^2 - (x_{t+1,j} - x_j)^2}{2\eta_{t,j}} + \sum_{t=1}^{T}\sum_{j=1}^{d}\frac{\eta_{t,j}}{2}\left(\frac{\partial f}{\partial x_{t,j}}(\boldsymbol{x}_t)\right)^2$$

$$\le \sum_{j=1}^{d}\sum_{t=1}^{T} \frac{(x_{t,j} - x_j)^2}{2}\left(\frac{1}{\eta_{t,j}} - \frac{1}{\eta_{t-1,j}}\right) + \sum_{j=1}^{d}\sum_{t=1}^{T}\frac{\eta_{t,j}}{2}\left(\frac{\partial f}{\partial x_{t,j}}(\boldsymbol{x}_t)\right)^2$$

$$\le \frac{D_{\infty}^2}{2\eta}\sum_{j=1}^{d}\sum_{t=1}^{T}\left(\sqrt{\sum_{i=1}^{t}\left(\frac{\partial f}{\partial x_{i,j}}(\boldsymbol{x}_i)\right)^2} - \sqrt{\sum_{i=1}^{t-1}\left(\frac{\partial f}{\partial x_{i,j}}(\boldsymbol{x}_i)\right)^2}\right) + \sum_{j=1}^{d}\sum_{t=1}^{T}\frac{\eta}{2\sqrt{\sum_{i=1}^{t}\left(\frac{\partial f}{\partial x_{i,j}}(\boldsymbol{x}_i)\right)^2}}\left(\frac{\partial f}{\partial x_{t,j}}(\boldsymbol{x}_t)\right)^2$$

$$\le \sum_{j=1}^{d}\left(\frac{D_{\infty}^2}{2\eta}\sqrt{\sum_{t=1}^{T}\left(\frac{\partial f}{\partial x_{t,j}}(\boldsymbol{x}_t)\right)^2} + \eta\sqrt{\sum_{t=1}^{T}\left(\frac{\partial f}{\partial x_{t,j}}(\boldsymbol{x}_t)\right)^2}\right)$$

$$= \sum_{j=1}^{d}\sqrt{2D_{\infty}^2\sum_{t=1}^{T}\left(\frac{\partial f}{\partial x_{t,j}}(\boldsymbol{x}_t)\right)^2}$$

$$\le \sqrt{2dD_{\infty}^2\sum_{t=1}^{T}\sum_{j=1}^{d}\left(\frac{\partial f}{\partial x_{t,j}}(\boldsymbol{x}_t)\right)^2}$$

$$= \sqrt{2dD_{\infty}^2\sum_{t=1}^{T}\|\nabla f(\boldsymbol{x}_t))\|^2} .$$

where the first inequality is by convexity, the second one by the projection lemma as the projection onto a hypercube equals performing the projection independently for each coordinate, the fifth one by Lemma 5 in (McMahan & Streeter, 2010), and the last one by the concavity of $\sqrt{\cdot}$. $\quad\square$

## C  THE HISTOGRAMS OF THE MAGNITUDE OF EACH UPDATE COORDINATE DURING THE ENTIRE TRAINING PHASE

In this section, we report the histograms of the absolute value of updates of Adam-$\ell_2$ vs. AdamW of all coordinates divided by $\alpha$ during the whole training process. From the figures shown below, we can clearly see that AdamW's updates remain in a much more concentrated scale range than Adam-$\ell_2$ during the entire training. Moreover, as the depth of the network grows, Adam-$\ell_2$'s updates become more and more dispersed, while AdamW's updates are still concentrated. *(Note that the leftmost bin contains all values equal or less than $2^{-27} \approx 10^{-8.1}$ and the rightmost bin contains all values equal to or larger than $1$.)*

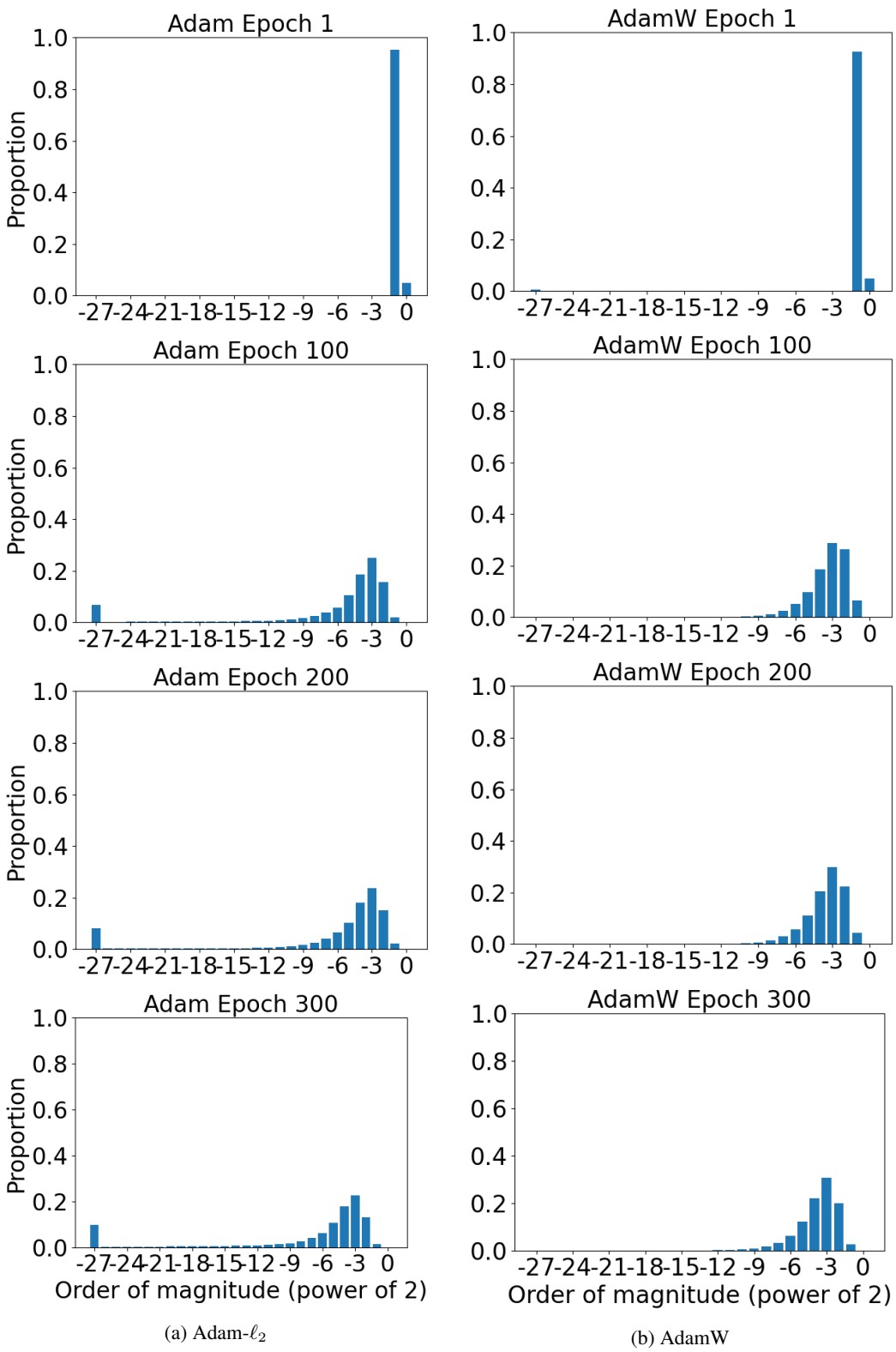

(a) Adam-$\ell_2$

(b) AdamW

Figure 7: The histograms of the magnitudes of all updates (without $\alpha$) of a 20-layer Resnet with BN removed trained by AdamW or Adam-$\ell_2$ on CIFAR10.

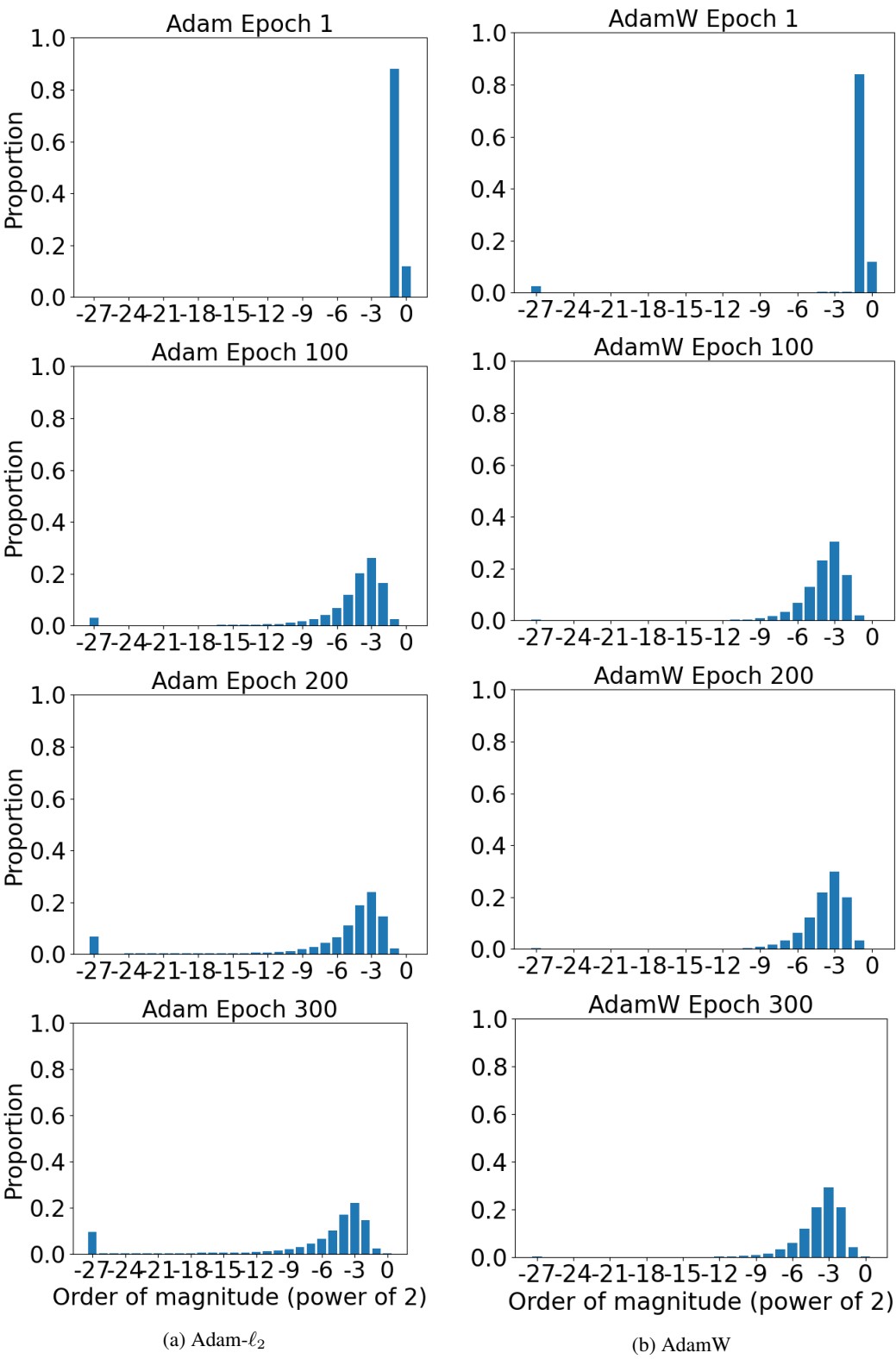

(a) Adam-$\ell_2$

(b) AdamW

Figure 8: The histograms of the magnitudes of all updates (without $\alpha$) of a 44-layer Resnet with BN removed trained by AdamW or Adam-$\ell_2$ on CIFAR10.

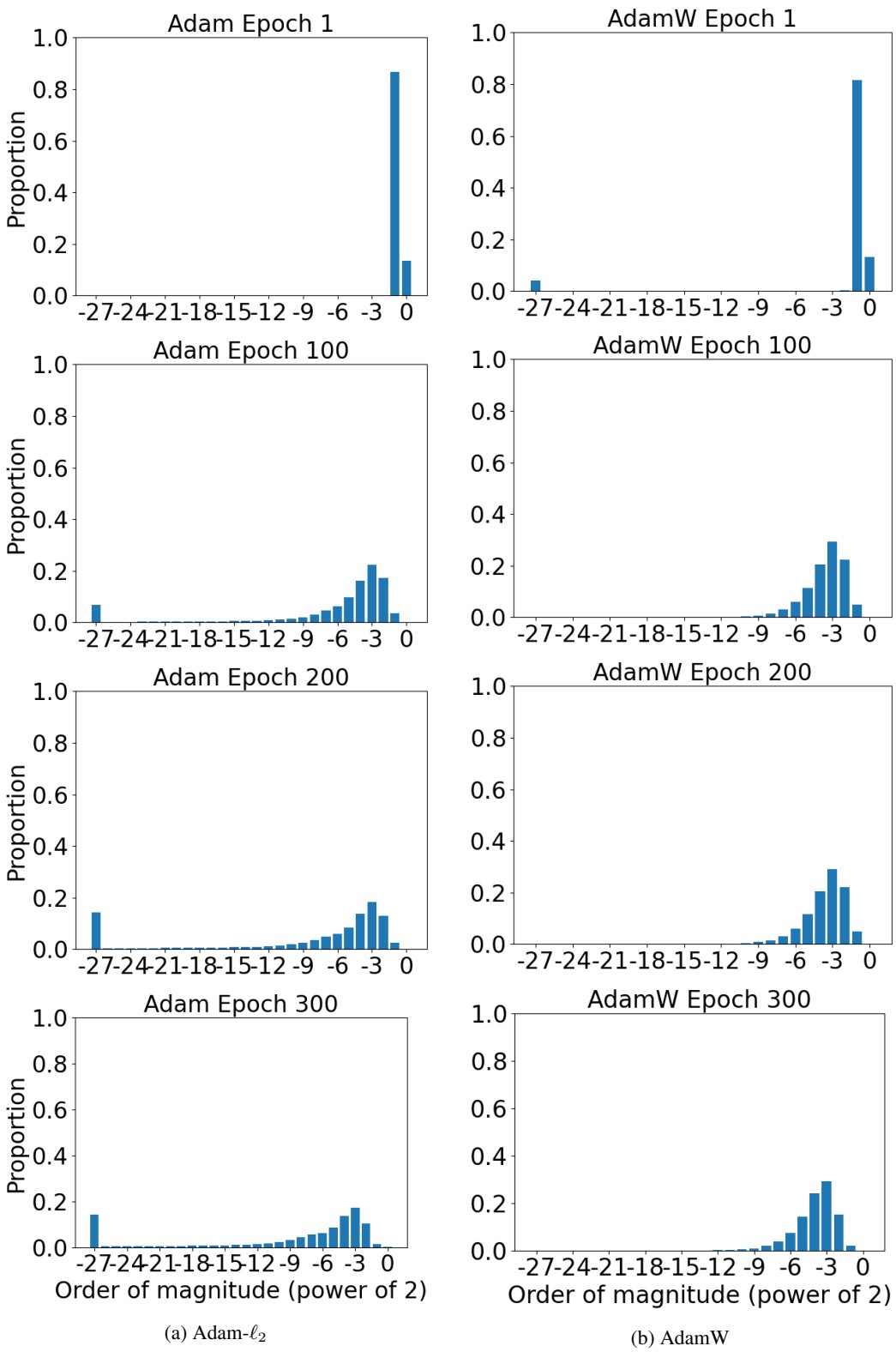

(a) Adam-$\ell_2$

(b) AdamW

Figure 9: The histograms of the magnitudes of all updates (without $\alpha$) of a 56-layer Resnet with BN removed trained by AdamW or Adam-$\ell_2$ on CIFAR10.

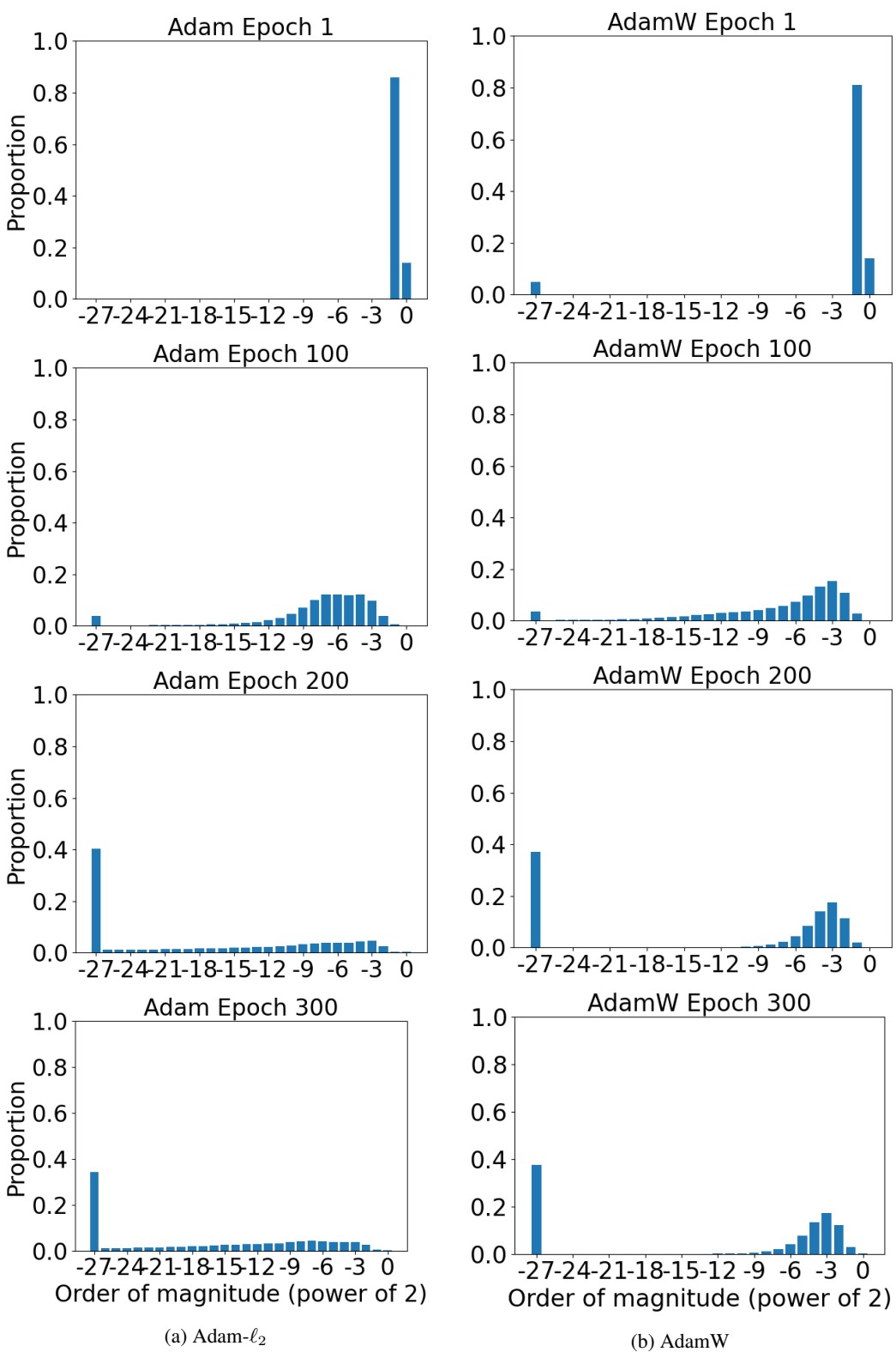

(a) Adam-$\ell_2$

(b) AdamW

Figure 10: The histograms of the magnitudes of all updates (without $\alpha$) of a 110-layer Resnet with BN removed trained by AdamW or Adam-$\ell_2$ on CIFAR10.

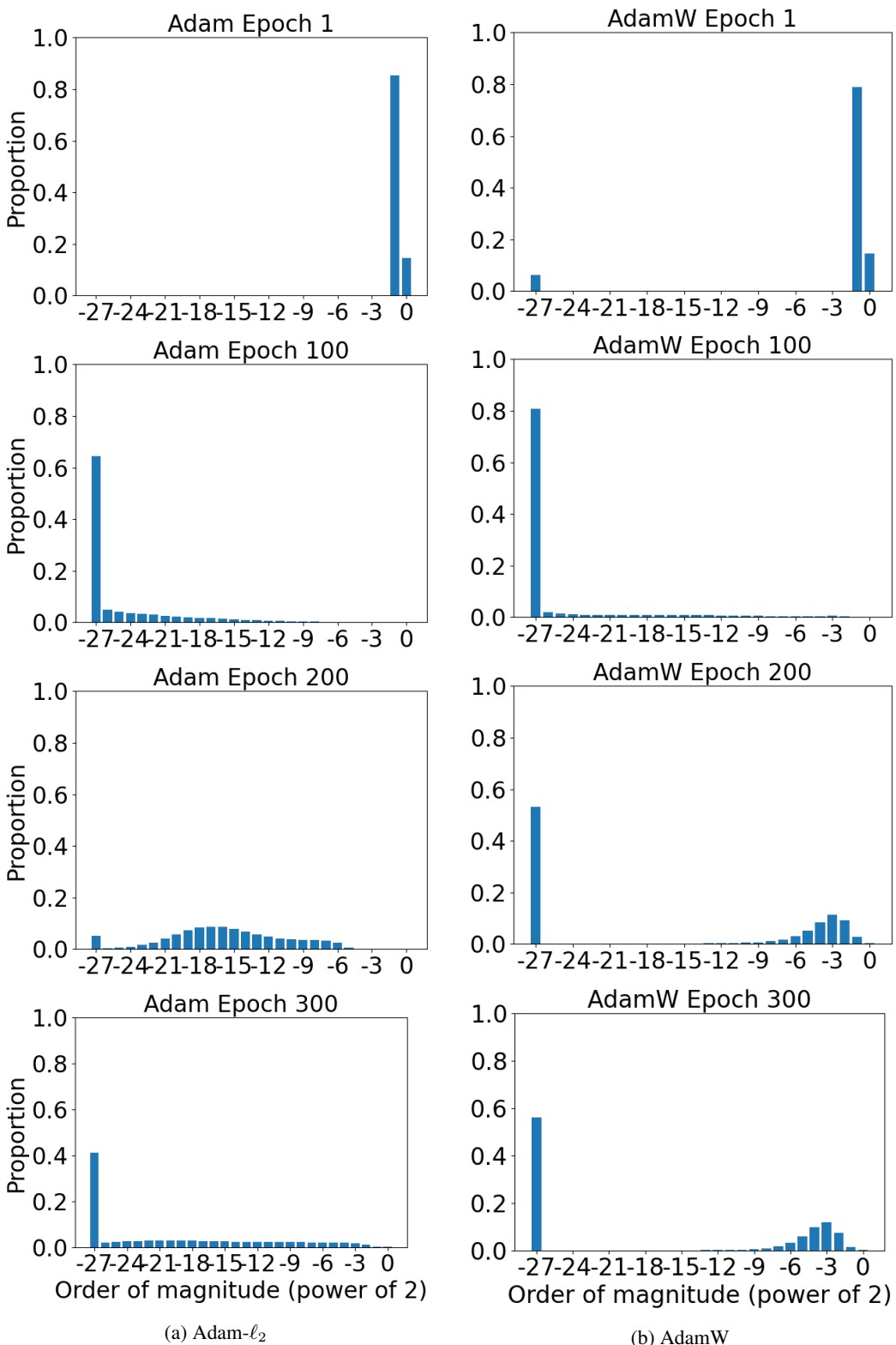

(a) Adam-$\ell_2$

(b) AdamW

Figure 11: The histograms of the magnitudes of all updates (without $\alpha$) of a 218-layer Resnet with BN removed trained by AdamW or Adam-$\ell_2$ on CIFAR10.

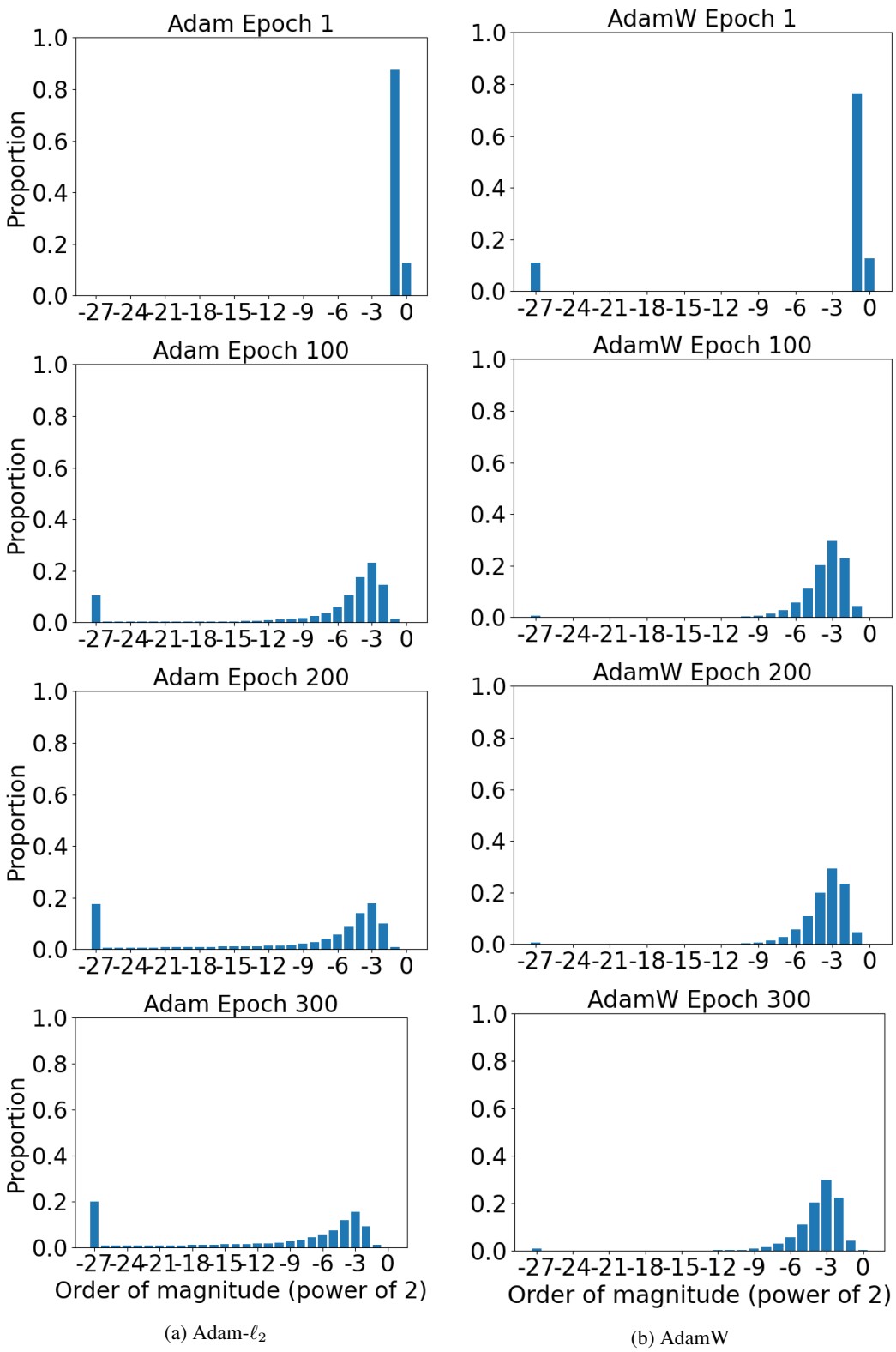

(a) Adam-$\ell_2$

(b) AdamW

Figure 12: The histograms of the magnitudes of all updates (without $\alpha$) of a 100-layer DenseNet-BC with BN removed trained by AdamW or Adam-$\ell_2$ on CIFAR100.

