# OpenReview forum: "Understanding AdamW through Proximal Methods and Scale-Freeness"
_ICLR.cc/2022/Conference — ICLR 2022 Submitted_

### Official Review · Reviewer_rZ6j · 2021-10-25

**Correctness:** 4
**Technical Novelty And Significance:** 4
**Empirical Novelty And Significance:** 3
**Recommendation:** 6
**Confidence:** 4

**Main Review:**

Strengths:
1. Authors show that AdamW can be seen as an approximation of the proximal updates, utilizing the closed form of proximal mapping of the regularizer. It is nice to see the reasonable theoretical explanations to the AdamW optimizer.
2. Authors analyze the scale-freeness property of AdamW, and further show the advantages of the scale-freeness property which is convincing to explain why AdamW gets better performance when the network is deeper or not equipped with batch normalization.

Weakness:
Authors conduct experiments in small-scale datasets with convolutional neural networks. Actually, a more realistic setting is to validate the vision transformer-based architectures, which is more meaningful because (1) transformer is equipped with AdamW as a standard setting (2)  vision transformers are not equipped with the batch normalization. So I thought if the theoretical analysis can be validated in the transformer-based architectures, and with larger dataset such as ImageNet, the results will be more convincing.

Typos: In section 4 the last sentence of the first paragraph, "??" should be modified.

**Summary Of The Paper:**

The paper give an analysis to the AdamW optimizer. This paper gives a theoretical view of the AdamW optimizing algorithm and connect it to the proximal gradients methods. Authors also explore in depth the advantages of AdamW over AdaM with l2 regularizer, from the perspective of scale-freeness property. By analyzing the scale-freeness property, authors find that the training is more stable in certain scenarios with large conditional numbers. Authors conduct experiments to validate their conclusions.

**Summary Of The Review:**

I lean to accept the paper. The paper analyze the AdamW optimizer well, and the experiments on small dataset support their conclusions. It will be more convincing to conduct experiments in more realistic scenarios such as transformer-based architectures trained on ImageNet.

---

> ### Author Response · Authors · 2021-11-13
> **Reply**
>
> Thank you for the positive comments and for pointing out the vision transformer-based architectures which is indeed a good scenario to validate our findings. We would be glad to explore this direction as future work.

---

### Official Review · Reviewer_tVBU · 2021-10-30

**Correctness:** 2
**Technical Novelty And Significance:** 2
**Empirical Novelty And Significance:** 2
**Recommendation:** 6
**Confidence:** 4

**Main Review:**


Strengths:
- Their work is well-motivated. An insightful interpretation or explanation of AdamW can benefit the whole deep/machine learning community in a way of guiding people on what scenario to use it and how.

Weakness & major concerns:
- The manuscript definitely need robust editing by a native speaker. To me sometimes it is really hard to follow.
- The math is not self-contained and the logical flow is disconnected. For example, the line above eq(3) "This results in the following update rule ...". But I don't think it is trivial to derive it from previous formulations. TBH, I fail to see how to derive eq(3). I would appreciate if the author could help me to see the connections between eq(3) and previous formulations.
- I assume eq(5) is the core updating rule of AdamProx, since there is no description of your algorithm. Are you claim that when $\eta_t$ is small enough, then AdamW is an approximation of eq(5)? Is this assumption too strong? When $\eta$ is too small, then the overall actual learning rate is almost zero which results in ineffective update.
- I feel the experiment section just carried out a comparison between AdamW and Adam-$\ell$2. It supposes to prodive more experiments with AdamProx.


**Summary Of The Paper:**

In this paper, the authors first interpret AdamW as an approximation of proximal mapping, and then proposed their own algorithm AdamProx. They aim to unravel the connection between AdamW and AdamProx from both theoretical and empirical perspective. They delicately designed some probing experiments to verify their hypothesis.

**Summary Of The Review:**

I recommend to reject the paper.

---

> ### Author Response · Authors · 2021-11-13
> **Reply**
>
> Thank you for considering our work “well-motivated” and seeing its potential benefits to “the whole deep/machine learning community in a way of guiding people on what scenario to use it and how”.
> We want to clarify a few points below:
> 1. **You wrote “then proposed their own algorithm AdamProx” in the summary and “I feel the experiment section just carried out a comparison between AdamW and Adam-2. It supposes to provide more experiments with AdamProx”.**
> > A: It seems we have failed to convey our message: The whole paper is devoted to AdamW, it was never our intention to advocate for use of AdamProx instead.
> >
> > Though we did write out the AdamProx update, we certainly do not claim it is superior to AdamW: instead, AdamProx is used to empirically validate our theoretical insight that AdamW is computing a classical proximal update. Thus, if AdamProx significantly outperformed (or underperformed) AdamW, this would actually be counter to our theoretical claim!
>
> 2. **I assume eq(5) is the core updating rule of AdamProx, since there is no description of your algorithm. Are you claim that when \eta_t  is small enough, then AdamW is an approximation of eq(5)? Is this assumption too strong? When \eta_t  is too small, then the overall actual learning rate is almost zero which results in ineffective update.**
> > (See also the first paragraph in red on Page 4 starting with “The careful reader might notice” and Section 4.3 on Page 8.)
> >
> > A: The reviewer is right that the approximation of AdamW to AdamProx does become less accurate when $\eta_t$ becomes too large, but in practice, $\eta_t$ is never too large. First, the argument relating AdamProx to AdamW is through a first-order Taylor expansion in $\eta_t$. The remainder term of this approximation is $O(\lambda \eta_t^2)$, which in practice we should always expect to be small.
> >
> > Beyond this theoretical argument, however, we did verify empirically that the approximation is good. In fact, in Figure 5, we empirically showed the case when $\eta_t$ = 1 for all $t$ meaning a constant scheduler. In such cases, AdamW and AdamProx still behave very similarly. We argue that this suffices to show that AdamW approximates AdamProx well enough for most practical learning rate schedules, e.g., cosine, exponential, polynomial, and step decay, as they all decrease from $\eta_0 = 1$.
>
> 3. **To derive Equation (3)**
> > A: We consider $\lambda > 0$ (written below Eq (1)) and $M_t = \eta_tI_d$ (written below Eq(2)) for $\eta_t > 0$. Then the right hand part inside the $\arg\min$ of the equation immediately above Eq (3) is a (unconstrained) strongly convex function of $x$. The minimum of such a function is the unique point where the gradient is $0$.
> >
> > We thus compute the gradient to get $\lambda x + p_t + M_t^{-1}(x - x_{t-1})$.
> >
> > We then solve the equation $\lambda x + p_t + M_t^{-1}(x - x_{t-1}) = 0$.
> >
> > As $M_t$ is positive definite, we left multiply both sides by $M_t$ to get $\lambda M_t x + M_t p_t + I_d(x - x_{t-1}) = 0$.
> >
> > Rearrange terms we have $(I_d + \lambda M_t)x = x_{t-1} - M_t p_t$ which gives us the solution
> >
> > $x_t = (I_d + \lambda M_t)^{-1}(x_{t-1} - M_t p_t)$
> >
> > as $I_d + \lambda M_t$ is still a positive definite matrix and thus invertible.
> >
> > We hope this clarifies the derivation of Equation (3).

---

> > ### Comment · Reviewer_tVBU · 2021-11-22
> > **Thanks for the authors' responses**
> >
> > I appreciate all the efforts the authors have devoted to improve their work. I would like to thank the authors' responses for addressing my concerns, which also helps me to have better understanding of their work.
> >
> > My further concerns are:
> > 1. To highlight the scale-freeness of AdamW, when compared with SGD or Adam-$\ell_2$, the authors propose to remove BN. I think this perhaps is a proper way to show the advantage of one method. But in practice, there is no reason to not use BN. I understand the main purpose of removing BN is to show the scale-freeness of AdamW. I just doubt whether this claim "the advantage of AdamW could be due to the scale-free updates" is still reasonable or not, when other algorithms equipped with BN.
> > 2. If the main work is to reveal the connection between proximal update and AdamW, I feel the novelty or contribution is limited. I think that would be great, if the authors can propose some insightful guidance for improve the algorithm even we don't restrict the other algorithms to use BN.
> >
> > For the above concerns, I only can increate my score from 3 to 5.

---

> > > ### Author Response · Authors · 2021-11-23
> > > **Thank you for your additional feedback!**
> > >
> > > Thank you very much for the additional feedback!
> > >
> > > We would like to address your further concerns below:
> > > 1. **To highlight the scale-freeness of AdamW, when compared with SGD or Adam-$\ell_2$, the authors propose to remove BN. I think this perhaps is a proper way to show the advantage of one method. But in practice, there is no reason to not use BN. I understand the main purpose of removing BN is to show the scale-freeness of AdamW. I just doubt whether this claim "the advantage of AdamW could be due to the scale-free updates" is still reasonable or not, when other algorithms equipped with BN.**
> > > > (Also see new changes on Page 2 in the second paragraph sentences in red color starting with “Note that the setting of removing BN”)
> > > >
> > > > A: The reasons we consider the setting of removing BN are manifold:
> > > >
> > > > First, we indeed want to isolate its effects on the scales of gradients. This was certainly our primary motivation.
> > > >
> > > > Secondarily, however, there are actually many situations in which one might prefer to not use BN in practice, for which we list some reasons below (for a more complete list, please refer to [Brock et al., ICML 2021]):
> > > >
> > > > 1. BN could incur significant memory overhead [Bulò et. al, CVPR 2018] and increase the training time [Gitman & Ginsburg, arXiv 2017].
> > > >
> > > > 2. When using BN, the model behaves differently during training and inference, causing an undesirable discrepancy [Summers & Dinneen, arXiv 2019; Singh & Shrivastava, CVPR 2019].
> > > >
> > > > 3. BN breaks the independence between training examples in the minibatch.
> > > >
> > > > Consequently, BN is not suitable for many cases, including distributed computing with a small minibatch per GPU [Wu & He, ECCV 2018, Goyal et al., arXiv 2017], sequential modeling tasks [Ba et al., arXiv 2016], and contrastive learning algorithms [Chen et al., ICML 2020].
> > > >
> > > > Considering these, Brock et al. [ICML 2021] even “anticipate that in the long term BN is likely to impede progress” and call “to identify a simple alternative”. Indeed, there is already active research in removing BN [De & Smith, NeurIPS 2020, Zhang et al. ICLR 2019], and there are already SOTA architectures that do not use BN including the Vision transformer pointed out by Reviewer rZ6j and the BERT model.
> > > >
> > > > Thus, we can conclude that the setting of removing BN is very much worth investigating. As an example, it gives insights on designing future algorithms optimizing models without BN, hinting that they should be scale-free.
> > >
> > > 2. **If the main work is to reveal the connection between proximal update and AdamW, I feel the novelty or contribution is limited. I think that would be great, if the authors can propose some insightful guidance for improve the algorithm even we don't restrict the other algorithms to use BN.**
> > > > A: As we added in the revised paper (see the second paragraph in red on Page 4 starting with “While perhaps not widely known”), this connection between the theory of **proximal methods**, a deep and beautiful branch of optimization, and **AdamW**, a very popular practical algorithm which still lacks a clear explanation of why it works so well, opens the door to understanding AdamW and to new ways to design optimization algorithms.
> > > >
> > > > So, we argue that finding such a connection has its own value because it provides a scientific explanation to a widely used heuristic. Moreover, this connection could bring new exciting algorithms that merge ideas from proximal methods in deep learning optimization algorithms. We do list some of these directions in Section 5, but way more could be discovered in the future. Just as the ICLR reviewer guideline puts it: “the research community is so big that somebody will find some value in the paper (maybe even a few years down the road), even if you don’t see it right now”.
> > > >
> > > > A good example is the AdaGrad algorithm [Duchi et al., COLT 2010] which was originally proposed for the online convex optimization setting. However, it inspired the development of Adam [Kingma & Ba, ICLR 2015] which became incredibly popular in optimizing deep neural networks.
> > > >
> > > > Moreover, we believe that if we understand how an algorithm works, then we can build upon it; otherwise, the only hope of improving it is through a semi-blind trial-and-error procedure. We find this resonates with your original comment that “an insightful interpretation or explanation of AdamW could benefit the whole deep/machine learning community in a way of guiding people on what scenario to use it and how”.

---

> > > > ### Comment · Reviewer_tVBU · 2021-11-29
> > > > **Thanks for addressing my most concerns**
> > > >
> > > > Since the authors addressed most of my concerns, I increased my score to 6.

---

> > > ### Author Response · Authors · 2021-11-27
> > > **Feedback request**
> > >
> > > Dear reviewer:
> > >
> > > We really appreciate your constructive feedback and your further concerns. We are glad that our initial response addressed your concerns and that you decided to increase the score. We have further answered your concerns on why removing BN is practical and reasonable, and why we feel so enthusiastic about discovering the connection between AdamW and the proximal methods and how it can be beneficial. Given that the deadline of discussion is really close, we would really like to learn whether you are satisfied with our further response or if you have any further comments? Thank you again and looking forward to hearing from you!

---

### Official Review · Reviewer_Vxer · 2021-11-02

**Correctness:** 3
**Technical Novelty And Significance:** 1
**Empirical Novelty And Significance:** 1
**Recommendation:** 3
**Confidence:** 4

**Main Review:**

Strengths:
(1)	It interprets AdamW as an approximation of a proximal gradient method, and shows the difference between AdamW and Adam-L2.
(2)	The authors also show that AdamW is “scale-freeness” algorithm which may enjoys faster convergence speed

Weaknesses:
(1)	In the abstract, the authors aim to solve the problem: why AdamW generalizes better than Adam-L2. However, in the paper, they actually did not solve or try to solve this problem.

(1.1)	They show the difference between AdamW and Adam-L2 from an approximation of a proximal gradient method. But they do not explain why different approximations give different generalizations? Why the approximation of AdamW is better than Adam-L2 in terms of generalization.

(1.2)	For the second contribution, AdamW is a “scale-freeness” algorithm that may enjoy faster convergence speed. First convergence speed does not have a necessary correlation to generalization, since one can converge to a very sharp minimum and may generalize worse. Secondly, For the “scale-freeness” algorithm, it is only possible to achieve faster convergence rate. But for AdamW, the authors do not show how/why adamw can achieve a faster convergence rate. This is a big gap.

(1.3) adam is  “scale-freeness”, but its performance is usually worse than adam-l2. So how “scale-freeness” explains this?

(2)	For interpreting AdamW as an approximation of a proximal gradient method and showing the difference between AdamW and Adam-L2, this seems to be very trivial in my view. What is the novelty or contribution of this?


**Summary Of The Paper:**

This work first re-interprets AdamW as an approximation of a proximal gradient method, which takes advantage of the closed-form proximal mapping of the regularizer instead of only utilizing its gradient information as in Adam-l2.  Then it further shows that AdamW is “scale-freeness” algorithm which may enjoys faster convergence speed.  In this way, this work may explain why AdamW generalizes better than Adam-L2.

**Summary Of The Review:**

Overall, this work does not provide very new insights for improving adamw or showing the better generalization of adamw over adam-l2. Moreover, it also does not have new proof techniques/frameworks.

---

> ### Author Response · Authors · 2021-11-13
> **Reply**
>
> We thank the reviewer for the feedback and would like to clarify the following points:
> 1. **In the abstract, the authors aim to solve the problem: why AdamW generalizes better than Adam-L2. However, in the paper, they actually did not solve or try to solve this problem.**
> > (See also the modified paper in Page 2 first paragraph, Page 2 first paragraph of Related Work Section, and Page 5 the red sentence in the second paragraph.)
> >
> > A: There is a really big misunderstanding here: We never tried to investigate the generalization ability of AdamW (and we have modified the paper to emphasize this point). We probably were not clear enough in our writing.
> >
> > We only mentioned that “to improve generalization, Adam is typically used in tandem with a squared $\ell_2$ regularizer” and “by enforcing the magnitude of the model weights to be small, weight decay has long been a standard technique to improve the generalization ability in machine learning”. These phrases are simply meant to reference the already known effect of weight decay (Krogh & Hertz, 1991; Bos & Chug, 1996).
> >
> > The rest of the paper does not mention the generalization ability at all.
> >
> > Again, we apologize for this misunderstanding and we have changed the paper to remedy it.
>
> 2. **Adam is “scale-freeness”, but its performance is usually worse than adam-l2. So how “scale-freeness” explains this?**
> > (See also the paragraph in red in the middle of Page 4.)
> >
> > A: Again, this seems to be a misunderstanding of our claims. We are sorry we were not able to properly convey our main message. Let us try to explain here in a better way: We do not study the generalization of optimizers. Instead, we study how an optimizer behaves when, in order to have a better generalization, an $\ell_2$ regularizer is used. That is, we assume that the $\ell_2$ regularizer will improve generalization, and instead wonder how it affects the optimizer performance.
> >
> > **In this specific case you mention, our only claim is that the loss of scale-freeness damages the performance.** So, Adam differs from Adam-$\ell_2$ as Adam-$\ell_2$ employs weight decay. Weight decay is of course well-known to improve performance, and so in this case it seems likely that the benefits of weight decay outstrip the penalty for losing the scale-freeness property.
> >
> > Thus, we cannot use scale-freeness alone to explain the comparison between Adam and Adam-$\ell_2$ because of the confounding factor of weight decay. However, when both Adam-$\ell_2$ and AdamW use weight decay, AdamW surpasses Adam-$\ell_2$ (Figure 3). This correlates with the hypothesis that the scale-freeness property is beneficial.
>
> 3. **For interpreting AdamW as an approximation of a proximal gradient method and showing the difference between AdamW and Adam-L2, this seems to be very trivial in my view. What is the novelty or contribution of this? Overall, this work does not provide very new insights for improving adamw or showing the better generalization of adamw over adam-l2. Moreover, it also does not have new proof techniques/frameworks.**
> > (See also the second paragraph in red on Page 4 starting with “While perhaps not widely known”.)
> >
> > A: While not widely known, the theory of proximal methods is a deep and beautiful branch of
> optimization *(see Proximal Algorithms (Parikh & Boyd, 2014) for a survey)*. On the other hand, even if AdamW is widely used, e.g., in training BERT (Devlin et al., 2019) and vision transformer-based architectures as Reviewer rZ6j pointed out, it still does not have any explanation.
> >
> > So, we might be wrong, but connecting all the plethora of results on proximal methods with AdamW seems extremely exciting to us. It opens the way to new ways to design optimization algorithms. Also, to our delight, we find the other reviewers consider our theoretical explanations of AdamW interesting (Reviewer xwd7), nice and reasonable (Reviewer rZ6j), and well-motivated and can benefit the whole deep/machine learning community (Reviewer tVBU).

---

> > ### Author Response · Authors · 2021-11-27
> > **Feedback request**
> >
> > Dear reviewer:
> >
> > We really appreciate your constructive feedback and have taken the effort on addressing your concerns and incorporating your suggestions into the paper, specifically on clarifying that our paper does not focus on the generalization ability of AdamW, why Adam is scale-free but inferior to Adam-$\ell_2$, and how the connection between AdamW and the proximal methods is non-trivial. Given that the deadline of discussion is really close, we would really like to learn whether you are satisfied with our response or if you have any further comments? Thank you again and looking forward to hearing from you!

---

> > ### Comment · Reviewer_Vxer · 2021-11-27
> > **Reply**
> >
> > (1)  The authors claim that they do not aim to analyze the generalization in this work. In this way, I do not get their target: they want to explain the advantages of AdamW. So what do these advantages refer to? is it not the better generalization ability? But their abstract claims that "However, even better performance can be obtained with AdamW, which decouples the gradient of the regularizer from the update rule of Adam-L2. Yet, we are still lacking a complete explanation of the advantages of AdamW." The advantages likely refer to "generalization".
> >
> > (2) For “scale-freeness”,  in response they claim "our only claim is that the loss of scale-freeness damages the performance." I do not understand how they have this kind of conclusion. Does the performance refer to "convergence" or "generalization"?
> >
> > (2.1) If performance refers to "convergence", the authors do not show how/why adamw can achieve a faster convergence rate since they only show that  AdamW is a “scale-freeness” algorithm that \textit{may} enjoy faster convergence speed.   For the “scale-freeness” algorithm, it is only possible to achieve a faster convergence rate. But for AdamW, the authors do not show how/why adamw can achieve a faster convergence rate. This is a big gap. What is more, Adam is “scale-freeness”, then does it mean Adam is faster than Adam-L2?
> >
> > (2.2)  If performance refers to "generalization",    “scale-freeness” does not have a necessary correlation to generalization. At least they do not prove this kind of result.
> >
> > (3) AdamW is a “scale-freeness” algorithm that \textit{may} enjoy faster convergence speed. The authors seem to transfer a possible event to an event that really happens.  But as mentioned in (2) they do not prove this kind of result.

---

> > > ### Author Response · Authors · 2021-11-28
> > > **Response to your further concerns**
> > >
> > > We are very displeased with the confrontational tone of the reviewer. This should be a conversation to clarify misunderstandings. Instead, the reviewer’s only defends their original (and incorrect) interpretation of our results.
> > >
> > > That said, the reviewer certainly must understand that *no* optimization algorithm can guarantee generalization in *all* machine learning scenarios. Indeed, as we all know, good generalization is *primarily* influenced by the network topology and the regularization, and *secondarily* influenced by the optimization algorithm. The optimization algorithm can “bias” towards some solutions instead of others, but its role is clearly model-dependent. In other words, it is possible to design networks that give poor generalization performance with *any* optimization algorithm. However, it should go without saying that poor performance in training can easily translate into a poor test performance, simply because we didn’t minimize the regularized objective function well enough (notice, for example, that in the well-known study [Wilson et al., NIPS 2017], train performance is actually usually correlated with test performance, and when it is not, (e.g. figure 2d), the relative difference in generalization is roughly $1\\%$). Here, as we clearly say in the red part in the related work section, we assume that the regularization and the topology of the network guarantee good generalization performance and we study algorithms from the perspective of convergence rate.
> > >
> > > [Wilson et al., NIPS 2017] Wilson, A.C., Roelofs, R., Stern, M., Srebro, N., & Recht, B. (2017). The Marginal Value of Adaptive Gradient Methods in Machine Learning. 31st Conference on Neural Information Processing Systems.]
> > >
> > > (1). **The authors claim that they do not aim to analyze the generalization in this work. In this way, I do not get their target: they want to explain the advantages of AdamW. So what do these advantages refer to? is it not the better generalization ability? But their abstract claims that "However, even better performance can be obtained with AdamW, which decouples the gradient of the regularizer from the update rule of Adam-L2. Yet, we are still lacking a complete explanation of the advantages of AdamW." The advantages likely refer to "generalization".**
> > > > A: No, they do not. They refer to convergence on the training objective, which was made clear in the response. We have also revised the paper and written it out in several places that we *do not* consider the generalization ability, see for example the modified paper on Page 2 first paragraph, Page 2 first paragraph of Related Work Section. The “advantages” here specifically refer to the better behavior in the training phase, which, as explained above and in the related work section, can translate to a smaller testing error in the presence of a network and regularization that guarantees good testing performance. Now, we would like to anticipate a possible additional question: how can we know if the assumption that the network and regularizer are chosen to generalize is true? We do know we are in this situation because we consider the testing error of all the algorithms only with their best regularization setting.
> > >
> > > (2). **For “scale-freeness”, in response they claim "our only claim is that the loss of scale-freeness damages the performance." I do not understand how they have this kind of conclusion. Does the performance refer to "convergence" or "generalization"?**
> > >
> > > (2.1). **If performance refers to "convergence", the authors do not show ... Adam is “scale-freeness”, then does it mean Adam is faster than Adam-L2?**
> > >
> > > (2.2). **If performance refers to "generalization", “scale-freeness” does not have a necessary correlation to generalization. At least they do not prove this kind of result.**
> > >
> > > (3). **AdamW is a “scale-freeness” algorithm that *may* enjoy faster convergence speed. The authors seem to transfer a possible event to an event that really happens. But as mentioned in (2) they do not prove this kind of result.**
> > > > A: First of all, as the reviewer admits, we do show theoretically that scale-freeness implies a possible improvement in convergence rate. However, it should go without saying, that only a *possible* improvement can be shown because in the worst-case the rates are optimal already. We are very surprised anyone would even raise this objection. This is exactly why more and more theory people are moving from worst-case guarantees to data-dependent ones.
> > > >
> > > > That said, the reviewer seems to have missed that we also do show empirically that deep learning objective functions give rise *exactly* to the conditions in which scale-freeness helps. Specifically, see Figure 3 in Section 4 that exactly shows the spread in the scales of the updates when without the scale-freeness property. Scale-free algorithms, by definition, are exactly insensitive to this spread and, by the theorem in section 3, they are equivalent to using the best diagonal preconditioner.

---

### Official Review · Reviewer_xwd7 · 2021-11-09

**Correctness:** 3
**Technical Novelty And Significance:** 3
**Empirical Novelty And Significance:** Not applicable
**Recommendation:** 6
**Confidence:** 4

**Main Review:**

Strengths:
1. The main contribution comes from providing the connection between the proximal operator and the AdamW updating, which is interesting.
2. The submission further shows that scale-freeness is an essential property of AdamW compared to Adam-l2. Moreover, scale-freeness can help mitigate the impact of condition number, which significantly affects the convergence speed.

Weaknesses:
1. One problem is that the submission shows that scale-freeness is essential for the optimizer. However, many optimizers share this property, such as adaGrad; this submission does not show why AdamW is generally better than the others
2. For Line 5 in Alg.1, if we scale the whole gradient $g_t$, we also get the scale-freeness? Do the author mean only scale the gradient of $f$. ie.e,$\nabla f$? It seems that all the adam-type optimizers all have this property. Why is AdamW special?

Minor Problem:
1. 'AdaGrad does not compute Hessians, and there is no reason to believe it approximates them in general.', which confused me. The quasi-Newton method also does not compute Hessians, but it indeed approximates the inv of Hessians. Moreover, scale-freeness certainly is superior in the scenario that the identity matrix can reasonably approximate the hessian. Why deny such a view here?

**Summary Of The Paper:**

An interesting finding shows that the advantages of AdamW compared to the vanilla Adam-l2.

**Summary Of The Review:**

An interesting paper provides a good finding.

---

> ### Author Response · Authors · 2021-11-13
> **Reply**
>
> We thank the reviewer for liking our paper and would like to address your concerns below:
>
> 1. **For Line 5 in Alg.1, if we scale the whole gradient g_t, we also get the scale-freeness? Does the author mean only scale the gradient of f. i.e., \nabla f? It seems that all the adam-type optimizers all have this property. Why is AdamW special?**
> > A: Indeed, our definition of scale-freeness specifically applies to the scaling of the gradient of $f$ (NOT the whole gradient including the regularization term). We also agree that the updates of Adam-$\ell_2$ will be invariant to scaling the whole gradient if we include the regularization term. However, we argue that our definition of scale-free is a more useful property:
> >
> > Our definition considers the case when different coordinates of gradients have very different scales as is the case in training deep neural networks without proper normalization techniques. Scale-free algorithms according to our definition are robust to such scenarios.
> >
> > In contrast, the scale of the regularization term depends on two terms: the user-specified value of $\lambda$, and the size of the weight matrices. The former is constant and so clearly does not exhibit a varying scale. The latter is likely also relatively constant across layers when using standard initialization schemes. This means that including the regularization term in the gradient when defining the scale-free property is not able to properly capture the true variation in the gradient magnitudes produced by the specific architecture.
>
> 2. **One problem is that the submission shows that scale-freeness is essential for the optimizer. However, many optimizers share this property, such as AdaGrad; this submission does not show why AdamW is generally better than the others**
> > (See also middle of Page 4 red paragraph starting with “We want to emphasize”)
> >
> > A: Not exactly: The presence of an $\ell_2$ regularizer makes AdaGrad and all other similar algorithms not scale-free. Without weight decay, AdaGrad is indeed scale-free just as the Adam-$\ell_2$ does when $\lambda = 0$. However, when employing the weight decay strategy same as Adam-$\ell_2$, namely adding the gradient of the $\ell_2$ regularizer directly to the gradient of $f$, as implemented in Tensorflow and Pytorch, AdaGrad is not scale-free anymore when using non-zero weight decay.
> >
> > Indeed, this is exactly the main point of our paper: Scale-freeness helps optimizers in deep learning and AdamW preserves this scale-freeness property even with an $\ell_2$ regularizer. We believe that the fact that this point is so easily missed is exactly why our paper is important.
>
> 3. **'AdaGrad does not compute Hessians, and there is no reason to believe it approximates them in general.', which confused me. The quasi-Newton method also does not compute Hessians, but it indeed approximates the inv of Hessians. Moreover, scale-freeness certainly is superior in the scenario that the identity matrix can reasonably approximate the hessian. Why deny such a view here?**
> > A: There is a key difference: Quasi-Newton methods actively try to estimate the inverse of the Hessian of any function through finite-time differences of gradients. AdaGrad instead uses the square root of the diagonal of the outer product of the gradients. Of course, there are functions in which this matrix approximates the Hessian, but it is a very small class of functions and in general, AdaGrad uses a matrix that is arbitrarily far from the Hessian.
> >
> > The idea that AdaGrad approximates the Hessian is a very common misconception.
> A better view is that, at least in the linear case, AdaGrad is adapting to the geometry of the variance in the data, not the geometry of the loss surface.
> >
> > That said, the reviewer is right that the case that “the identity matrix can reasonably approximate the hessian” would give scale-freeness a big advantage; however, even in the case of Hessians being close to identity matrices, AdaGrad’s updates still do not contain any second-order derivative information.

---

> > ### Author Response · Authors · 2021-11-27
> > **Feedback request**
> >
> > Dear reviewer:
> >
> > We really appreciate your constructive feedback and have taken the effort on addressing your concerns and incorporating your suggestions into the paper, specifically on our scale-freeness v.s. scaling the whole gradient, when AdaGrad is scale-free and when it is not, and why we claim that AdaGrad does not approximate/compute the hessian. Given that the deadline of discussion is really close, we would really like to learn whether you are satisfied with our response or if you have any further comments? Thank you again and looking forward to hearing from you!

---

### Author Response · Authors · 2021-11-13
**We have revised the paper, thanks to the constructive feedback from the reviewers.**

We thank the reviewers for the feedback and questions. In particular, we realized that our writing was suboptimal because some reviewers seem to have confused our main claims. We have now revised the paper to sharpen our message (see changes highlighted in red). Given the misunderstandings, we hope the reviewers will take the time to re-read the paper and revise their reviews.

In addition, per the suggestion of Reviewer tVBU, we have requested one of our co-authors who is a native speaker to carefully proofread the whole manuscript once again and we hope it now has a much better presentation.

Also note that due to the space limit, we have moved the proof of Theorem 1 to Appendix A.

---

### Author Response · Authors · 2021-11-20
**Feedback request**

Dear reviewers:

We really appreciate your constructive feedback and have updated the submission and posted our answers to your concerns. Given that the deadline of the second stage of discussion is really close but we have yet to receive your further response, we would like to hear from you on whether you are satisfied with our updates? If so, could you please re-evaluate your decisions; if not, could you let us know your further concerns?

Thank you so much!

---

### Decision · Program_Chairs · 2022-01-20

**Decision:**

Reject

**Comment:**

I agree that reviewer Vxer was confrontational and abusive, especially in the response to the author's rebuttal, and believe that some form of sanction or reprimand is appropriate. That said, I do think that "performance" should be evaluated on both convergence rate and generalization.  Figure 3 does suggest some improvement on generalization for deep versions of resnet without batch normalization.

The three less offensive reviewers all indicated weak acceptance.  One reviewer pointed out the weakness of only getting positive results on deep versions of resnet with batch normalization removed.  Results on transformers, where Adam is typically used, would be more compelling.  This is my primary issue with the paper.  It has not demonstrated improvement in the standard practice of resnet (with batch normalization) and has not presented experiments on transformers.  The theoretical analysis is not aimed at explaining why the improvement is only observed on deep resnets with batch normalization removed or why L2 regularization seems to be of no value when batch normalization is present.  I understand that these are very difficult questions.

The paper has no champion and I am personally concerned about the significance of the contribution.